# Pairing of holes by confining strings in antiferromagnets

F. Grusdt,[1, 2, *] E. Demler,[3] and A. Bohrdt[4, 5]

[1]*Department of Physics and Arnold Sommerfeld Center for Theoretical Physics (ASC),*
*Ludwig-Maximilians-Universität München, Theresienstr. 37, München D-80333, Germany*
[2]*Munich Center for Quantum Science and Technology (MCQST), Schellingstr. 4, D-80799 München, Germany*
[3]*Institut für Theoretische Physik, ETH Zurich, 8093 Zurich, Switzerland*
[4]*ITAMP, Harvard-Smithsonian Center for Astrophysics, Cambridge, MA 02138, USA*
[5]*Department of Physics, Harvard University, Cambridge, Massachusetts 02138, USA*
(Dated: October 6, 2022)

In strongly correlated quantum materials, the behavior of charge carriers is dominated by strong electron-electron interactions. These can lead to insulating states with spin order, and upon doping to competing ordered states including unconventional superconductivity. The underlying pairing mechanism remains poorly understood however, even in strongly simplified theoretical models. Recent advances in quantum simulation allow to study pairing in paradigmatic settings, e.g. in the $t - J$ and $t - J_z$ Hamiltonians. Even there, the most basic properties of paired states of only two dopants, such as their dispersion relation and excitation spectra, remain poorly studied in many cases. Here we provide new analytical insights into a possible string-based pairing mechanism of mobile holes in an antiferromagnet. We analyze an effective model of partons connected by a confining string and calculate the spectral properties of bound states. Our model is equally relevant for understanding Hubbard-Mott excitons consisting of a bound doublon-hole pair or confined states of dynamical matter in lattice gauge theories, which motivates our study of different parton statistics. Although an accurate semi-analytic estimation of binding energies is challenging, our theory provides a detailed understanding of the internal structure of pairs. For example, in a range of settings we predict heavy states of immobile pairs with flat-band dispersions – including for the lowest-energy $d$-wave pair of fermions. Our findings shed new light on the long-standing question about the origin of pairing and competing orders in high-temperature superconductors.

## I. INTRODUCTION

The history of physics has repeatedly taught us that nature tends to realize richer structures than one might first suggest. The most important example, by far, is constituted by the theory of atoms which has evolved from Thomson's featureless plum pudding model to our current picture of precisely quantized energy shells and a nucleus with structures down to the level of individual quarks. The only way to reveal such structures is to perform increasingly more precise measurements, or, with the benefit of hindsight and a microscopic Hamiltonian at hand, perform increasingly more precise numerical simulations. Here we address the question how much, and which, structure paired charge carriers in correlated quantum matter may have.

To date, important aspects of strongly correlated electrons remain poorly understood. Part of the reason is that the nature and microscopic structure of the emergent charge carriers is not fully understood. Among the most famous puzzles is the origin of pairing in high-temperature superconductors [1, 2], but similar questions arise in heavy-fermion compounds [3], organic superconductors [4] and most recently twisted bilayer graphene [5]. However the problems are not limited to paired states of matter: the nature of charge carriers in the exotic normal phases of these strongly correlated systems is likewise debated and subject of active research.

Remarkably, the prevailing picture of quantum matter is one with more-or-less featureless charge carriers, with little or no rigid internal structure taken into account in calculations. Some theoretical approaches assume fractionalization of quantum numbers, which leads to rich and interesting physics, but short-range spatial fluctuations are rarely considered in detail. Among the reasons for this restriction is the difficulty to directly experimentally visualize spatial structures of quickly fluctuating charges. While ultracold atoms in optical lattices [6, 7] have taken very promising steps in this direction [8, 9], they too have not managed to fully visualize the internal structure of doped charge carriers yet. Another reason may be the lasting influence of Anderson's RVB theory of high-$T_c$ superconductivity [10], which assumes point-like charge carriers moving in a surrounding spin-liquid – in some sense the antithesis of any theory assuming spatial structures of charge carriers.

On the other hand, it is not for a lack of ideas which kinds of internal structures could emerge: Even before the discovery of high-$T_c$ superconductors, Bulaevksii et al. proposed the existence of string-like states with internal vibrational excitations [11], a view taken up by Brinkman and Rice to understand dynamical properties of charge carriers [12]; in 1988, Trugman applied this idea to pairs of holes [13] and in the same year Shraiman and Siggia analyzed two-hole string states in greater detail [14]. Their conclusions at the time were mixed: while they found mechanisms supporting pairing, they also identified unfavorable effects such as frustration of the

* Corresponding author email: fabian.grusdt@lmu.de

pair-kinetic energy due to the underlying Fermi-statistics of the holes. Today these works stand out as being among the few and first attempting a description starting from the strong coupling antiferromagnetic (AFM) Mott limit (large Hubbard-$U$). But it appears that the community then focused more on approaches inspired by the weak-coupling (small Hubbard-$U$) limit of the theoretical models [2], which more naturally led to the magnon-exchange picture [15]. There, magnetic fluctuations provide the glue between point-like charge carriers, and the theoretical framework shares more similarities with the successful BCS theory of conventional superconductors. As almost all ideas in the field, this picture has been debated [16].

Nevertheless, over time the idea that charge carriers have a pronounced spatial structure was reclaimed several times. In 1996, Laughlin and co-workers proposed a phenomenological parton theory of doped holes, including a confining linear string tension [17, 18]; different kinds of spatial strings, termed phase-strings, were introduced in 1996 by Weng and co-workers [19, 20], and their effect on pairing was recently analyzed [21]; signatures for the more traditional $S^z$-string fluctuations were reported in large-scale DMRG simulations by White and Affleck in 2001 [22]; in 2007 Manousakis proposed a string-based interpretation of one-hole ARPES spectra and in the same work envisioned a pairing mechanism of holes constantly exchanging fluctuating strings [23]; in 2013 exact numerical simulations by Vidmar et al. in truncated bases, closely related to the string picture, have also revealed signatures for pairs with a rich spatial structure [24]; already in 2000 and 2001 large-scale Monte Carlo simulations by Brunner et al. [25] and Mishchenko et al. [26] have revealed long-lived vibrational excitations of individual doped holes; and in the past few years, the present authors have added new evidence for the existence of long-lived rotational and vibrational string states of individual charge carriers [27–29]; Vibrational peaks have also long been known to exist and recently further confirmed in linear-spin wave models of doped holes [30–33]. Finally, Hubbard-Mott excitons formed by a bound pair of a doublon and a hole [34] have been proposed to have a rich internal structure [35 **?** ].

In this article, we revisit the idea that mobile dopants, the charge carriers in doped AFM Mott insulators, can form bound states with a rich spatial structure. Specifically, we derive a semi-analytical theory of pairs of holes connected by a confining string which fluctuates only through the motion of the charges at its ends. Our approach is very similar in spirit to the much earlier work by Shraiman and Siggia mentioned above [14]; in fact we confirm several of their predictions and discuss them in the context of three decades worth of new results including from far advanced numerics. This includes one of their most exciting – though often overlooked – prediction that two fermionic holes can form infinitely heavy pairs with a flat-band dispersion at very low energies. In fact, we find that that these flat bands have $d$-wave character. This result sheds new light on the wealth of competing ordered states observed in cuprates and numerically found in the closely related $t - J$ and Fermi-Hubbard models [36–38].

Back-to-back with this article, we are publishing a separate work focusing on a numerical analysis of rotational two-hole spectra in the $t - J_z$ and $t - J$ models [39]. There we achieve full momentum resolution on extended four-leg cylinders, and compare our numerical results to the semi-analytic calculations performed in the present article. The focus of the present article is on the semi-analytical method itself, including its formal derivation. Moreover, the calculations we perform here are applicable in a larger class of models: as explained below, our main assumption is that two partons on a square lattice are connected by a rigid string $\Sigma$ which creates a memory of the parton's motion in the wavefunction. In the case of mobile holes doped into an AFM Mott insulator, the string directly encodes the prevalent spin-charge correlations. But similar situations can be found in lattice gauge theories in a strongly confining regime where the dynamics of the corresponding electric field strings is dominated by fast charge fluctuations.

The goal of our present work is primarily to understand the properties of pairs of mobile dopants, i.e. their spatial structure and energy spectrum, including their rotational quantum numbers and effective mass. Previously much emphasis has been on the question whether the dopants pair up; i.e. about the magnitude and sign of the binding energy

$$E_{\mathrm{bdg}} = 2(E_1 - E_0) - (E_2 - E_0) \qquad (1)$$

where $E_n$ is the ground state energy in the presence of $n$ dopants. While we also view this as an important issue, we argue that it is often not a well-suited question for a semi-analytical approach, since it strongly depends on details; e.g. addressing it requires precise knowledge of the one-hole energy. In this article we take the view that the structure of the paired state can be very different from the structure of one-dopant states. Our goal is to understand the former, and we leave the question of how and when excited (or ground) states of pairs can decay into individual single-dopant states to future analysis (only a brief discussion within our model will be provided). Moreover, we note that the question of pairing is not identical to a question about the existence superconductivity: instead of condensing into a superconductor, pairs of holes may also crystalize and form a pair-density wave at finite doping [40].

Nevertheless, we will address the origins of pairing within our semi-analytical framework. To this end we identify competing effects which tend to increase or decrease the binding energy. Our results shed new light on earlier predictions [13, 14]: (i) fermionic statistics of the dopants frustrates the pair's kinetic energy, which is unfavorable for pairing; (ii) the most detrimental effect for pairing comes from the hard-core property of two dopants, which cannot occupy the same site; Moreover, we reveal (iii) a geometric spinon-chargon repul-

sion in dimensions $d \geq 2$ which enhances the one-dopant energy $E_1$ [41] and thus favors pairing. In addition to all of these, further contributions stemming from low-energy spin-fluctuations in the background are expected, whose quantitative effects are more challenging to predict and will thus be left to future work to explore. Finally, we note that in specifically tailored settings our simplifying assumptions become quantitatively accurate: In Ref. [42] we proposed a mixed-dimensional bilayer model with strong rung singlets where we demonstrated strong string-based pairing of doped holes with a binding energy scaling as $E_{\mathrm{bdg}} \simeq t^{1/3}J^{2/3}$, when the hole tunneling $t \gg J$ exceeds the rung super-exchange $J$. Recently, in a closely related mixed-dimensional two-leg ladder, ultracold fermions directly observed strong hole binding [43], realizing a decades old toy model of pairing [44, 45]. The internal structure of hole pairs in these models can also be described by the theoretical model we develop here.

This article is organized as follows. In Sec. II we briefly discuss the microscopic models motivating our analysis. Sec. III constitutes the main body of our article: there we develop the effective string model to describe bound states of two mobile partons connected by a strongly confining string. Our focus is on two holes, but as we discuss in detail in Appendix A, the formalism we develop is also applicable to pairs of more general partons, in particular spinon-chargon pairs [29]. In the second main part of the article, Sec. IV, we present results from our analytical formalism and discuss possible implications for general pairing mechanisms in doped AFMs. We close with a summary and outlook in Sec. V.

## II.  MICROSCOPIC MODELS

In this article we introduce and solve an effective theory describing bound states of holes. As described in detail in Sec. III, we will make approximations on the level of both the Hilbert space and the Hamiltonian. Nevertheless, our starting point are microscopic models of doped AFM Mott insulators to which, we argue, our results apply within some approximations. Critical minds should simply view these models as motivating our effective theory, although our numerical analysis in Ref. [39] indicates remarkable similarities with the semi-analytical predictions derived here.

The system most closely related to our effective theory is constituted by the 2D $t-J_z$ model on a square lattice, with Hamiltonian

$$\hat{\mathcal{H}}_{t-J_z} = -t\,\hat{\mathcal{P}}\sum_{\langle \boldsymbol{i},\boldsymbol{j}\rangle}\sum_{\sigma}\left(\hat{c}^{\dagger}_{\boldsymbol{i},\sigma}\hat{c}_{\boldsymbol{j},\sigma}+\mathrm{h.c.}\right)\hat{\mathcal{P}}+$$
$$+ J_z\sum_{\langle \boldsymbol{i},\boldsymbol{j}\rangle}\hat{S}^z_{\boldsymbol{i}}\hat{S}^z_{\boldsymbol{j}} - \frac{J_z}{4}\sum_{\langle \boldsymbol{i},\boldsymbol{j}\rangle}\hat{n}_{\boldsymbol{i}}\hat{n}_{\boldsymbol{j}}. \quad (2)$$

Here $\hat{c}_{\boldsymbol{j},\sigma}$ defines the underlying particles, with spin-index $\sigma = \uparrow,\downarrow$ and density $\hat{n}_{\boldsymbol{j}} = \sum_{\sigma}\hat{c}^{\dagger}_{\boldsymbol{j},\sigma}\hat{c}_{\boldsymbol{j},\sigma}$; the tunneling

FIG. 1. We work in an effective Hilbertspace consisting of pairs of dopants (red and blue) connected by a string $\Sigma$ on a square lattice. Every state $|\boldsymbol{x}_1, \Sigma\rangle$ avoiding double occupancies of any site with two dopants is associated with a unique state $|\Psi(\boldsymbol{x}_1, \Sigma)\rangle$ in the microscopic model. (a) A typical example with string length $\ell_{\Sigma} = 3$. (b) Rare loop configurations leading to double-occupancies of dopants have no correspondence in the microscopic model.

amplitude $t$ describes hopping of these particles, and $\hat{\mathcal{P}}$ projects on a sector with a given total number of doublons, holes and singly-occupied sites. $J_z > 0$ is an AFM Ising coupling between the spins $\hat{S}^z_{\boldsymbol{j}} = \sum_{\sigma}(-1)^{\sigma}\hat{c}^{\dagger}_{\boldsymbol{j},\sigma}\hat{c}_{\boldsymbol{j},\sigma}$ which we assume to be antiferromagnetic throughout.

In principle the Hamiltonian in Eq. (2) can be defined with particles $\hat{c}_{\boldsymbol{j},\sigma}$ of any exchange statistics, bosonic or fermionic. While this makes no difference for zero and one mobile dopant, the statistics plays an important role if two dopants of the same type are considered. Since the fermionic case is more closely related to the celebrated Fermi-Hubbard model with its AFM ground state at half filling, it usually takes center stage. However, we find it instructive to consider the bosonic version – without any direct connection to a Hubbard Hamiltonian – as well. In fact, quantum simulators using ultracold atoms have been proposed which allow the realization of both cases in experiments [46–49].

Likewise, we can consider the AFM $t-J$ model in a 2D square lattice with arbitrary underlying statistics. Its Hamiltonian is given by

$$\hat{\mathcal{H}}_{t-J} = -t\,\hat{\mathcal{P}}\sum_{\langle \boldsymbol{i},\boldsymbol{j}\rangle}\sum_{\sigma}\left(\hat{c}^{\dagger}_{\boldsymbol{i},\sigma}\hat{c}_{\boldsymbol{j},\sigma}+\mathrm{h.c.}\right)\hat{\mathcal{P}}+$$
$$+ J\sum_{\langle \boldsymbol{i},\boldsymbol{j}\rangle}\hat{\boldsymbol{S}}_{\boldsymbol{i}}\cdot\hat{\boldsymbol{S}}_{\boldsymbol{j}} - \frac{J}{4}\sum_{\langle \boldsymbol{i},\boldsymbol{j}\rangle}\hat{n}_{\boldsymbol{i}}\hat{n}_{\boldsymbol{j}}. \quad (3)$$

Now $J > 0$ is the strength of antiferromagnetic SU(2)-invariant Heisenberg interactions between spins $\hat{\boldsymbol{S}}_{\boldsymbol{j}} = \sum_{\alpha,\beta}\hat{c}^{\dagger}_{\boldsymbol{j},\alpha}\frac{1}{2}\boldsymbol{\sigma}_{\alpha\beta}\hat{c}_{\boldsymbol{j},\beta}$.

In both the $t-J$ and $t-J_z$ models, different types of dopants can be considered. The most often studied case, which is also our primary focus, constitutes pairs of two indistinguishable holes; owing to the particle-hole symmetry of the models, one can interchangeably consider two indistinguishable doublons however. A second, closely related case corresponds to Hubbard-Mott excitons [35], where pairs of doublons and holes can form. In this case, exchange statistics again plays no role on the level of the $t-J_{(z)}$ model, since the dopants define distinguishable conserved particles. Experimentally, all these

situations can be addressed by state-of-the-art ultracold atom experiments [6, 7].

## III.   EFFECTIVE STRING MODEL

In this section, we introduce an effective string model to describe tightly bound pairs of two holes. Our approach is motivated by considering two dopants moving in a Néel state, modeled by the 2D $t - J^z$ or $t - J$ model. To make analytical progress, we perform approximations on the effective Hilbert space (see Fig. 1) as well as the effective Hamiltonian. While we expect our approximate description to be most accurate for the $t - J^z$ model, it should also capture the essential physics of related models, such as the $t - J$ model, as long as charge fluctuations dominate, $t \gg J$, and they feature strong local AFM correlations at zero doping. In these cases, the geometric string approach, developed originally for single dopants, can be applied [8, 11, 12, 14, 27].

In the subsequent sections, we will always talk about paired holes and consider different kinds of statistics. However, as discussed in Sec. II, these can interchangeably be considered as general types of dopants, or even more generally as two partons.

### A.   String Hilbert space and effective Hamiltonian

In a perfect Néel state the motion of a hole leaves behind a string $\Sigma$ of displaced spins, which allows to associate distinct hole trajectories with orthogonal states $|\Psi(\Sigma)\rangle$ in the quantum many-body system (Fig. 1); here strings denote hole trajectories with all self-retracing components removed. Exceptions, where two different strings $\Sigma_1 \neq \Sigma_2$ correspond to identical many-body states $|\Psi(\Sigma_1)\rangle = |\Psi(\Sigma_2)\rangle$, are associated with so-called Trugman loops [13]. Since the number of Trugman loops is small compared to the exponentially growing number of string states (Tab. I), their effect is generally found to be small [13, 27]. Although in exceptional cases the small effect of loops can still dominate, e.g. for the very narrow center-of-mass dispersion of a single hole in the 2D $t - J^z$ model [13, 50], we neglect such loops in the following and study only the dominant effects of string formation. Loop effects can be re-introduced perturbatively in the end [27].

#### 1.   Hilbert space

While the set of string states $\{|\Psi(\Sigma)\rangle\}$ defines an overcomplete basis, we approximate our Hilbertspace and formally define a set of two-hole string states:

$$|\boldsymbol{x}_1, \Sigma\rangle, \qquad \boldsymbol{x}_1 \in \mathbb{Z}^2, \quad \Sigma \in \mathrm{BL}. \qquad (4)$$

Here $\boldsymbol{x}_1$ denotes the location of the first hole in the 2D square lattice, and $\Sigma$ is the string which connects $\boldsymbol{x}_1$ to

| $\ell_\Sigma$ | no. of states $d(\ell_\Sigma)$ | double-occupancies | Trugman loops |
|---|---|---|---|
| 1 | 4 | 0 | 0 |
| 2 | 12 | 0 | 0 |
| 3 | 36 | 0 | 0 |
| 4 | 108 | 7.4% | 0 |
| 5 | 324 | 0 | 0 |
| 6 | 972 | 2.5% | 0 |
| 7 | 2,916 | 0 | 0.55% |
| 8 | 8,748 | 0.91% | 1.3% |

TABLE I. **Imperfections of the string model.** The string basis with two distinguishable holes includes states $|\boldsymbol{x}_1, \Sigma\rangle$ corresponding to unphysical double-occupancies of holes in the associated microscopic states $|\Psi(\boldsymbol{x}_1, \Sigma)\rangle$. Their relative fraction of all string states $N(\ell_\Sigma)$ of a given length $\ell_\Sigma$ is indicated. The relative number of Trugman loop configurations [13] is also shown.

$\boldsymbol{x}_2 = \boldsymbol{x}_1 + \boldsymbol{R}_\Sigma$ at its opposite end (Fig. 1). The strings $\Sigma$ can be represented by the sites of a Bethe lattice (BL), or Cayley tree, with coordination number $z = 4$, see Fig. 2 (a). Similar to the construction of the celebrated Rokhsar-Kivelson quantum dimer model [51], we postulate that the new basis states are orthonormal,

$$\langle \boldsymbol{x}_1', \Sigma' | \boldsymbol{x}_1, \Sigma \rangle = \delta_{\Sigma, \Sigma'} \delta_{\boldsymbol{x}_1, \boldsymbol{x}_1'}. \qquad (5)$$

Every new basis state is associated with a unique microscopic two-hole state $|\Psi(\boldsymbol{x}_1, J^z)\rangle$ in the original $t - J^z$ or $t - J$ model. Some states in the new model describe unphysical double occupancies with holes: in this case the associated state becomes $|\Psi(\boldsymbol{x}_1, \Sigma)\rangle = 0$ (Fig. 1). The fraction of such strings is relatively small, however, and decreases with increasing string lengths (Tab. I).

So far we work in first quantization and assign separate labels to the two holes. Later (see III D) we will generalize our approach to situations with indistinguishable holes, with bosonic or fermionic statistics.

#### 2.   Effective Hamiltonian

Next we define the effective Hamiltonian $\hat{\mathcal{H}}^{\mathrm{eff}}$ in the approximated Hilbert space of the string model. To this end we require that the matrix elements satisfy:

$$\langle \boldsymbol{x}_1', \Sigma' | \hat{\mathcal{H}}^{\mathrm{eff}} | \boldsymbol{x}_1, \Sigma \rangle = \langle \Psi(\boldsymbol{x}_1', \Sigma') | \hat{\mathcal{H}} | \Psi(\boldsymbol{x}_1, \Sigma) \rangle, \qquad (6)$$

where $\hat{\mathcal{H}}$ corresponds to the respective microscopic model Hamiltonian ($t - J^z$, $t - J$,...).

As a result of the microscopic nearest-neighbor (NN) hopping $t_2$ of hole 2 at site $\boldsymbol{x}_2$ on the lattice, we obtain NN hopping on the Bethe lattice (Fig. 2 (b)):

$$\hat{\mathcal{H}}_{t,2}^{\mathrm{eff}} = t_2 \sum_{\boldsymbol{x}_1} |\boldsymbol{x}_1\rangle\langle\boldsymbol{x}_1| \otimes \sum_{\langle \Sigma', \Sigma \rangle} |\Sigma'\rangle\langle\Sigma| + \mathrm{h.c.} \qquad (7)$$

The NN hopping $t_1$ of hole 1 gives rise to a correlated NN tunneling of $\boldsymbol{x}_1$ and a simultaneous change of the

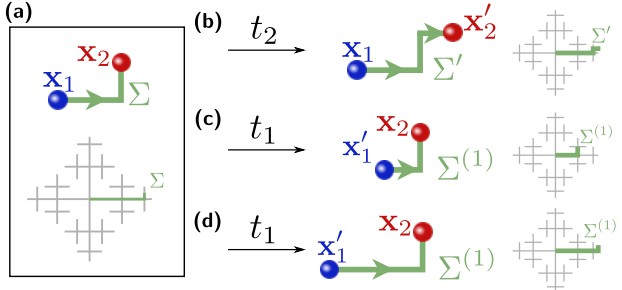

FIG. 2. The hopping part of the effective Hamiltonian $\hat{\mathcal{H}}_t^{\text{eff}}$ describes NN tunneling of holes 1 ($t_1$) and 2 ($t_2$) on the square lattice. The string $\Sigma$ from hole 1 to 2 changes accordingly. We illustrate a typical initial state (a), which is coupled to neighboring string states on the Bethe lattice by $t_2$ (b). The coupling $t_1$ creates a string state $\Sigma^{(1)}$ which is a further neighbor of $\Sigma$ on the Bethe lattice, either by re-tracing (b) or extending (d) the first string segment.

first string segment of $\Sigma \to \Sigma^{(1)}(\boldsymbol{x}_1', \boldsymbol{x}_1, \Sigma)$:

$$\hat{\mathcal{H}}_{t,1}^{\text{eff}} = t_1 \sum_{\langle \boldsymbol{x}_1', \boldsymbol{x}_1 \rangle} \sum_{\Sigma} |\boldsymbol{x}_1'\rangle\langle\boldsymbol{x}_1| \otimes |\Sigma^{(1)}(\boldsymbol{x}_1', \boldsymbol{x}_1, \Sigma)\rangle\langle\Sigma| + \text{h.c..}$$

(8)

If the first string segment in $\Sigma$ points along $\boldsymbol{x}_1' - \boldsymbol{x}_1$, it is removed to obtain $\Sigma^{(1)}$ (Fig. 2 (c)); otherwise, the string is extended by adding a new string segment pointing along $\boldsymbol{x}_1' - \boldsymbol{x}_1$ at the beginning of $\Sigma$ to obtain $\Sigma^{(1)}$ (Fig. 2 (d)).

Similarly, we obtain the potential terms $\hat{\mathcal{H}}_J^{\text{eff}}$ in the effective Hamiltonian. They do not change the positions $\boldsymbol{x}_{1,2}$ of the holes, and we neglect off-diagonal matrix elements (6) for which $\Sigma' \neq \Sigma$. Hence, formally we can write:

$$\hat{\mathcal{H}}_J^{\text{eff}} = \sum_{\boldsymbol{x}_1} \sum_{\Sigma} V_\Sigma |\boldsymbol{x}_1, \Sigma\rangle\langle\boldsymbol{x}_1, \Sigma|,$$

(9)

where $V_\Sigma$ is a function of $\Sigma$ on the Bethe lattice only. Note that Trugman loops correspond to local minima in the Bethe lattice potential $V_\Sigma$, which allows for a systematic tight-binding treatment of Trugman loops within our model so far, even when $t_{1,2} \gg J$ or $J_z$ [27].

The complete effective Hamiltonian we consider is

$$\hat{\mathcal{H}}^{\text{eff}} = \hat{\mathcal{H}}_{t,1}^{\text{eff}} + \hat{\mathcal{H}}_{t,2}^{\text{eff}} + \hat{\mathcal{H}}_J^{\text{eff}}.$$

(10)

### 3. Linear string approximation

Since the dimension of the string Hilbert space grows exponentially with the maximum string length $\ell_{\text{max}}$,

$$d_{\text{tot}}(\ell_{\text{max}}) = \sum_{\ell_\Sigma=1}^{\ell_{\text{max}}} \underbrace{4 \times 3^{\ell_\Sigma - 1}}_{=d(\ell_\Sigma)},$$

(11)

further approximations are required to make analytical progress. For a general string potential $V_\Sigma$, all states

in the string Hilbert space are coupled to each other by the tunneling terms. Next, by simplifying the potential $V_\Sigma$, many symmetry sectors emerge which are only weakly coupled and can be described by a simpler effective Hamiltonian that will be derived.

Since we are mostly interested in the regime $t \gg J$, we expect that inhomogeneities of $V_\Sigma$ on the scale of $J$ play a sub-dominant role. Such fluctuations of $V_\Sigma$ from string to string (or site to site on the Bethe lattice) result from string-string interactions [52], and appear like a weak disorder potential. We can include their effect on a mean-field level by averaging the potential over all strings of a given length:

$$V(\ell) = \frac{1}{d(\ell)} \sum_{\Sigma:\ell_\Sigma=\ell} V_\Sigma.$$

(12)

The resulting problem is highly symmetric since all branches on the Bethe lattice corresponding to the same string length are equivalent.

As a further approximation, we can estimate the string-length potential $V(\ell) \approx V_{\text{LST}}(\ell)$ by considering only straight strings in Eq. (12). Since string-string interactions are always attractive, this linear string theory (LST) estimate also defines an upper bound for the averaged potential:

$$V(\ell) \leq V_{\text{LST}}(\ell) = \frac{dE}{d\ell} \times (\ell_\Sigma - 1) + g_{\text{cc}}^{(0)} \delta_{\ell_\Sigma, 1} + \mu_{\text{cc}}.$$

(13)

Here $dE/d\ell$ denotes the linear string tension, $g_{\text{cc}}^{(0)}$ is a nearest-neighbor hole-hole interaction, and $\mu_{\text{cc}}$ an overall energy shift. The overall energy of the two holes is measured relative to the undoped parent antiferromagnetic state.

In the case of a microscopic $t - J^z$ model, we obtain:

$$\frac{dE}{d\ell} = J^z, \quad g_{\text{cc}}^{(0)} = -\frac{J^z}{2}, \quad \mu_{\text{cc}} = 4J^z.$$

(14)

More generally, we can derive the LST potential by applying the frozen spin approximation and expressing the potential in terms of local spin-spin correlations of the undoped parent antiferromagnet, see Refs. [27, 41]. For a doped $J_1 - J_2$ spin model on a square lattice, with NN hopping of the holes, this yields:

$$\frac{dE}{d\ell} = 2J_1 (C_2 - C_1) + 2J_2 (C_1 + C_4 - 2C_2),$$

(15)

$$g_{\text{cc}}^{(0)} = J_1 C_1 - \frac{J}{4},$$

(16)

$$\mu_{\text{cc}} = -8 \left( J_1 C_1 + J_2 C_2 - \frac{J}{4} \right).$$

(17)

Here $J_1$ and $J_2$ denote the NN and diagonal NNN Heisenberg couplings on the square lattice; $J$ denotes the strength of the local attraction $-J/4\hat{n}_i\hat{n}_j$ in Eq. (3) and is treated as an independent parameter here. The correlators are $C_1 = \langle \hat{\boldsymbol{S}}_i \hat{\boldsymbol{S}}_{i+e_x} \rangle$ (NN), $C_2 = \langle \hat{\boldsymbol{S}}_i \hat{\boldsymbol{S}}_{i+e_x+e_y} \rangle$ (NNN), $C_3 = \langle \hat{\boldsymbol{S}}_i \hat{\boldsymbol{S}}_{i+2e_x} \rangle$ and $C_4 = \langle \hat{\boldsymbol{S}}_i \hat{\boldsymbol{S}}_{i+2e_x+e_y} \rangle$; they depend on the ratio $J_1/J_2$ [53].

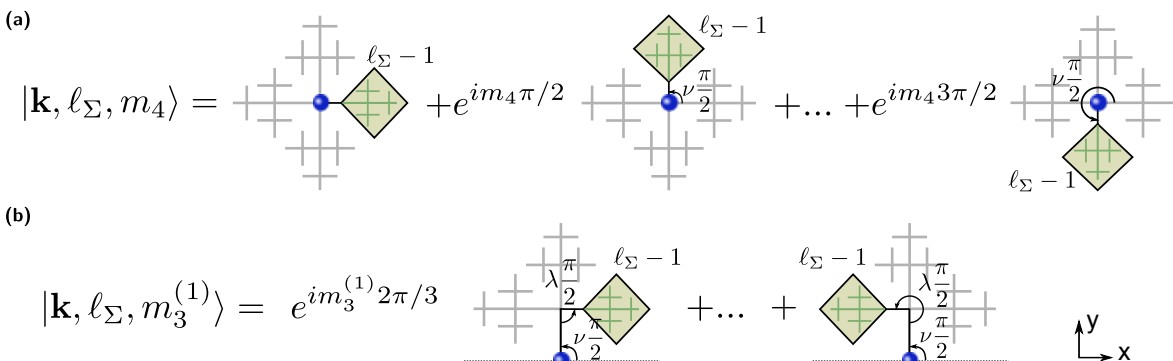

FIG. 3. The rotational basis consists of superpositions of string states on the Bethe lattice, defined around a rotational center. There is (a) one $m_4$ quantum number around the central site, and (b) one $m_3$ quantum number per every node except the center. Angles $\nu\pi/2$ are measured relative to the $x$-axis and $\lambda\pi/2$ relative to the preceding string segment. Note that in (b) only one branch of the Bethe lattice, with fixed $\nu$, is shown.

#### 4. Momentum basis

So war we have defined the string model in the real-space basis. Because of the overall translational symmetry of the model, $[\hat{\mathcal{H}}^{\text{eff}}, \hat{T}] = 0$ where the operator $\hat{T}$ translates both holes and the string by a discrete lattice vector, we can also work in the momentum-space basis. We define the following total momentum states,

$$|\boldsymbol{k}, \Sigma\rangle = \frac{1}{\sqrt{V}} \sum_{\boldsymbol{x}_1} e^{i\boldsymbol{k}\cdot\boldsymbol{x}_1} |\boldsymbol{x}_1, \Sigma\rangle, \qquad (18)$$

where $V = L^2$ denotes the total area of the square lattice.

By the symmetry, the effective Hamiltonian is block-diagonal, with entries

$$\hat{\mathcal{H}}^{\text{eff}}(\boldsymbol{k}) = \sum_{\Sigma,\Sigma'} \langle \boldsymbol{k}, \Sigma' | \hat{\mathcal{H}}^{\text{eff}} | \boldsymbol{k}, \Sigma\rangle \; |\Sigma'\rangle\langle\Sigma|, \qquad (19)$$

and we can calculate the bound state properties independently at different total momenta $\boldsymbol{k}$.

### B. Rotational excitations and truncated basis

In the simplified string potential $V(\ell_\Sigma)$ all strings of a given length are equivalent. This symmetry is conserved by the hopping term $t_2$ of the second hole, which leads to delocalization on the Bethe lattice. Hence, as long as the first hole cannot tunnel, i.e. for $t_1 = 0$, our problem with an exponentially large Hilbert space can be mapped to a single particle problem on a semi-infinite one-dimensional lattice $\ell_\Sigma = 1, 2, ...., \infty$ in a central potential $V(\ell_\Sigma)$, see [27]. This situation is of great relevance for describing spinon-chargon bound states in an effective string basis [27, 54], as will be discussed further in III C 1.

In general, the tunneling of the first hole, $\propto t_1$, will break the symmetry between the string states, because some couple to longer, others to shorter strings depending on their orientation. Nevertheless, we find it useful to

work in a basis of string states which takes into account the equivalence of strings when $t_1 = 0$: As we show later, for $C_4$-rotation invariant total momenta $\boldsymbol{k}^{\text{C4IM}}$ (C4IM), even $t_1 \neq 0$ keeps the symmetry intact. Away from the C4IM, $t_1$ introduces weak couplings between the symmetric eigenstates, which we will explicitly take into account.

The quantum numbers characterizing the symmetric model correspond to one discrete rotational eigenvalue for each node in the Bethe lattice [27]. At the first node, around the central site $\ell_\Sigma = 0$ in the Bethe lattice, one obtains a discrete $C_4$ rotational symmetry with eigenvalues $\exp(im_4\pi/2)$ where $m_4 = 0, 1, 2, 3$. The symmetric states with string length $\ell_\Sigma = 1$ are

$$|\boldsymbol{k}, \ell_\Sigma = 1, m_4\rangle = \frac{1}{2} \sum_{\nu=0}^{3} e^{im_4\nu\pi/2} |\boldsymbol{k}, \ell_\Sigma = 1, \nu\rangle, \qquad (20)$$

where $\nu\pi/2$ denotes the angle of the string segment $\Sigma$ with the $x$-axis, see Fig. 3 (a).

Each higher node, around sites on the Bethe lattice corresponding to string length $\ell \geq 1$, is associated with a discrete $C_3$ rotational symmetry. The corresponding eigenvalues are $\exp(im_3^{(\ell)} 2\pi/3)$, with $m_3^{(\ell)} = 0, 1, 2$. The corresponding symmetric states are defined as

$$|\boldsymbol{k}, \ell_\Sigma, m_3^{(\ell)}\rangle = \frac{1}{\sqrt{3}} \sum_{\lambda=1}^{3} e^{im_3^{(\ell)}\lambda 2\pi/3} |\boldsymbol{k}, \ell_\Sigma - 1, \lambda\rangle, \qquad (21)$$

where $\lambda\pi/2$ denotes the angle between the previous and last string segment, see Fig. 3 (b).

The most general states are defined by an entire set of angular momentum quantum numbers,

$$\boldsymbol{m} = (m_4, m_3^{(1)}, m_3^{(2)}, ...). \qquad (22)$$

I.e. instead of the original basis states $|\boldsymbol{x}, \Sigma\rangle$, we work in the basis: $\{|\boldsymbol{k}, \ell_\Sigma, \boldsymbol{m}_\Sigma\rangle\}$.

As a final step, we simplify our problem further by working with a truncated set of basis states. We discard non-trivial $m_3$ rotational excitations on nodes higher

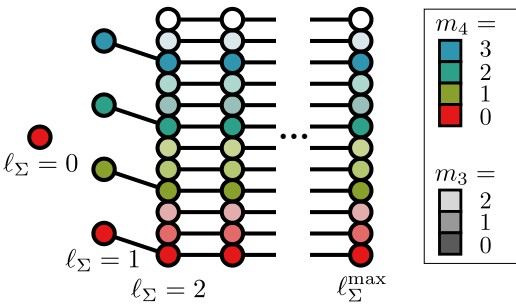

FIG. 4. The truncated basis consists of rotational states with $m_4$ and $m_3^{(1)} \equiv m_3$ quantum numbers only, defined around the first hole (asymmetric approximation). Later we will symmetrize this basis to treat both chargons on an equal footing. Solid lines indicate non-zero matrix elements which preserve the rotational symmetries in a linear string model. The state with $\ell_\Sigma = 0$ can be formally added, but should be removed to describe cases where two holes cannot occupy the same site.

than one, i.e. we only take into account the quantum number $m_3^{(1)}$ and set all $m_3^{(n)} = 0$ for $n \geq 2$. The truncated basis consists of the states

$$\{|\boldsymbol{k}, \ell_\Sigma, m_4, m_3^{(1)}\rangle\} \qquad \text{truncated basis} \qquad (23)$$

and allows us to work with very large cut-offs in the string length, see Fig. 4.

The main motivation for discarding higher rotational states is their higher energy in the purely symmetric limits (at $\boldsymbol{k}^{\text{C4IM}}$ or when $t_1 = 0$). By studying the importance of the $m_3^{(1)}$ states within our reduced basis, we obtain an estimate for the effect of higher rotational manifolds in general.

We note that for cases with distinguishable holes, or more generally distinguishable partons, one should place the heavier parton (corresponding to a weaker hopping element) in the center. Around this first hole, labeled by $n = 1$, the rotational states can then be defined. In the case of spinon-chargon pairs in the context of the $t - J$ [29] or $t - J_z$ model [27], which we discuss in more detail below, this is typically the spinon since $t < J_{(z)}$ [17].

We emphasize that for $t_1 = t_2$, by choosing to define rotational states around parton $n = 1$, the symmetry between the two partons is explicitly broken in the truncated basis; we will refer to this ansatz as the *asymmetric approximation*. Below, in Sec. III D, we will show how (anti-) symmetrization of both cases, with parton $n = 1$ and $n = 2$ in the center respectively, leads to a more accurate truncated basis in the case $t_1 = t_2$, going beyond the asymmetric approximation.

### C. Distinguishable partons: Mott-Hubbard excitons

Now we are in a position to further simplify the effective string problem, still assuming distinguishable holes

and working in the asymmetric approximation. The first step is to calculate all required matrix elements in Eq. (19) for the truncated basis from Eq. (23).

The string potential is completely diagonal, and we simply get

$$\langle \boldsymbol{k}, \ell'_\Sigma, m'_4, m'_3 | \hat{\mathcal{H}}_J^{\text{eff}} | \boldsymbol{k}, \ell_\Sigma, m_4, m_3 \rangle =$$
$$= V(\ell_\Sigma)\, \delta_{\ell'_\Sigma, \ell_\Sigma} \delta_{m'_4, m_4} \delta_{m'_3, m_3}. \quad (24)$$

The tunneling of the second hole, $\propto t_2$, is diagonal in the rotational quantum numbers by construction [27] but changes the string length by one unit. For $\ell_\Sigma \geq 2$ we obtain

$$\langle \boldsymbol{k}, \ell'_\Sigma, m'_4, m'_3 | \hat{\mathcal{H}}_{t,2}^{\text{eff}} | \boldsymbol{k}, \ell_\Sigma, m_4, m_3 \rangle =$$
$$= \sqrt{3} t_2 \left( \delta_{\ell'_\Sigma, \ell_\Sigma - 1} + \delta_{\ell'_\Sigma, \ell_\Sigma + 1} \right) \delta_{m'_4, m_4} \delta_{m'_3, m_3}, \quad (25)$$

where the pre-factor $\sqrt{3} = \sqrt{z-1}$ related to the coordination number $z = 4$ of the square lattice appears. The states with $\ell_\Sigma = 1$ only have an $m_4$ quantum number and couple to $\ell'_\Sigma = 2$ states, i.e.

$$\langle \boldsymbol{k}, \ell'_\Sigma, m'_4, m'_3 | \hat{\mathcal{H}}_{t,2}^{\text{eff}} | \boldsymbol{k}, 1, m_4 \rangle = \sqrt{3} t_2\, \delta_{\ell'_\Sigma, 2}\, \delta_{m'_4, m_4}\, \delta_{m'_3, 0}; \quad (26)$$

note that $\ell_\Sigma = 0$ states are not included here since we assume hard-core holes. Overall, we obtain for the hopping amplitude $J_-^{(2)}$ of the second hole from $\ell_\Sigma$ to $\ell_\Sigma - 1$,

$$J_-^{(2)}(\boldsymbol{k}; m'_4, m'_3; m_4, m_3) \equiv$$
$$\equiv \langle \boldsymbol{k}, \ell_\Sigma - 1, m'_4, m'_3 | \hat{\mathcal{H}}_{t,2}^{\text{eff}} | \boldsymbol{k}, \ell_\Sigma, m_4, m_3 \rangle =$$
$$= \sqrt{3} t_2\, \delta_{m'_4, m_4} \delta_{m'_3, m_3}, \quad (27)$$

and $J_+^{(2)} = (J_-^{(2)})^*$ for the reverse process, $\ell_\Sigma \to \ell_\Sigma + 1$.

More complicated matrix elements $\propto t_1$ are generated by the hopping of the first hole, which may change both angular momenta $m_4$ and $m_3$. We only need to calculate the matrix elements going from $\ell_\Sigma$ to $\ell'_\Sigma = \ell_\Sigma - 1$, which reduces the calculational workload since the final state must have $m'_3 = 0$. The remaining non-zero matrix elements $J_+^{(1)} = (J_-^{(1)})^*$ describe the reverse process, going from $\ell_\Sigma$ to $\ell'_\Sigma = \ell_\Sigma + 1$, and are directly obtained from the former by complex conjugation. We find from a detailed calculation

$$J_-^{(1)}(\boldsymbol{k}; m'_4, m'_3; m_4, m_3) \equiv$$
$$\equiv \langle \boldsymbol{k}, \ell_\Sigma - 1, m'_4, m'_3 | \hat{\mathcal{H}}_{t,1}^{\text{eff}} | \boldsymbol{k}, \ell_\Sigma, m_4, m_3 \rangle =$$
$$= \delta_{m'_3, 0} \frac{t_1}{4} \sum_{\nu=0}^{3} e^{i \frac{\pi}{2} \nu (m_4 - m'_4)} \chi_{m_4, m_3}(\nu, \boldsymbol{k}), \quad (28)$$

where we worked in the string configuration basis to obtain

$$\chi_{m_4, m_3}(\nu, \boldsymbol{k}) = \frac{e^{i \pi m_4}}{\sqrt{3}} \sum_{\nu'=1}^{3} e^{-i\nu' \frac{\pi}{2} m_4 + i \frac{2\pi}{3} m_3 \nu' - i \boldsymbol{k} \cdot \boldsymbol{e}_{\nu - \nu' + 2}}. \quad (29)$$

In the last expression, we defined the unit vector $\boldsymbol{e}_\lambda$ for $\lambda = 0, 1, 2, 3 \bmod 4$ to point along $\lambda \pi / 2$ relative to the $x$-axis, i.e. $\boldsymbol{e}_0 = \boldsymbol{e}_x$, $\boldsymbol{e}_1 = \boldsymbol{e}_y$, $\boldsymbol{e}_2 = -\boldsymbol{e}_x$ and $\boldsymbol{e}_3 = -\boldsymbol{e}_y$.

Summarizing, we obtain an effective Hamiltonian with the matrix elements calculated above,

$$
\hat{\mathcal{H}}^{\mathrm{eff}}(\boldsymbol{k}) = \sum_{\ell_\Sigma} \Bigg[ \sum_{\boldsymbol{m}} V(\ell_\Sigma) \, |\ell_\Sigma, \boldsymbol{m}\rangle\langle\ell_\Sigma, \boldsymbol{m}| +
$$
$$
+ \sum_{\boldsymbol{m}, \boldsymbol{m}'} \sum_{n=1,2} \Big( J_-^{(n)}(\ell_\Sigma; \boldsymbol{k}; \boldsymbol{m}'; \boldsymbol{m})
$$
$$
\times |\ell_\Sigma - 1, \boldsymbol{m}'\rangle\langle\ell_\Sigma, \boldsymbol{m}| + \mathrm{h.c.} \Big) \Bigg]. \quad (30)
$$

Here the summation over $\boldsymbol{m}$, $\boldsymbol{m}'$ is restricted to $m_4, m_3^{(1)}$ in our truncated basis; for later purposes we also formally included an $\ell_\Sigma$-dependence in $J_-^{(n)}$, although we found the expressions above to be independent of $\ell_\Sigma$. This Hamiltonian can be easily diagonalized using exact numerical techniques, with large cut-offs for the maximum string length $\ell_{\max}$.

### 1. Soft-core holes and spinon-chargon pairs

Before proceeding, we note that the formalism introduced above for distinguishable holes is very general and can be extended to describe string-bound states of any two different partons. Examples of particular relevance include strongly bound holes in mixed-dimensional bilayers [7, 42, 43] and one-hole spinon-chargon pairs [17, 27–29]. An accurate description of the latter is important, since their energy relative to the two-hole chargon-chargon string states determines whether tightly-bound pairs of holes are energetically favorable in an antiferromagnet.

In both examples we mentioned, the two partons may occupy the same site. While we already made an approximation and included states with two partons on one site in the effective string basis for string lengths $\ell_\Sigma > 0$, we have not included the $\ell_\Sigma = 0$ state with two partons on the same site and without a string. To treat more general parton models, the $\ell_\Sigma = 0$ state can be easily included in our formalism, however. This leaves the form of the effective Hamiltonian Eq. (30) unchanged, but the hopping elements now include couplings to $\ell_\Sigma = 0$ and the potential term $V(\ell_\Sigma = 0)|\ell_\Sigma = 0\rangle\langle\ell_\Sigma = 0|$ has to be added as well.

Still assuming that both partons $n = 1, 2$ can only tunnel between NN sites on the physical square lattice, we obtain hopping elements from $\ell_\Sigma = 1$ to $\ell_\Sigma = 0$. For the first, central parton $(n = 1)$ we get

$$
J_-^{(1)}(\boldsymbol{k}; m_4)|_{\ell_\Sigma=1 \to \ell_\Sigma=0} = \frac{t_1}{2} \sum_{\nu=0}^{3} e^{i\nu \frac{\pi}{2} m_4} e^{-i\boldsymbol{k}\cdot\boldsymbol{e}_\nu}. \quad (31)
$$

Note that only one rotational index $m_4$ appears since the state $|\boldsymbol{k}, \ell = 0\rangle$ has no rotational quantum numbers, and the states $|\boldsymbol{k}, \ell = 1, m_4\rangle$ at $\ell = 1$ have one $m_4$ number.

For the second, orbiting parton $(n = 2)$ we obtain

$$
J_-^{(2)}(\boldsymbol{k}; m_4)|_{\ell_\Sigma=1 \to \ell_\Sigma=0} = 2t_2 \delta_{m_4,0}. \quad (32)
$$

In contrast to the other hopping elements, see Eq. (27), this matrix element has strength $2 = \sqrt{4}$ instead of $\sqrt{3}$. This is a consequence of the enhanced connectivity of the $\ell_\Sigma = 0$ state, which couples to 4 instead of the usual 3 longer string states [11, 27].

Finally, for the spinon-chargon case in an antiferromagnet, it is natural to assume NNN hopping of the spinon due to spin-exchange processes in the Hamiltonian. An extension of our formalism to this case has been used in [29], and a self-contained derivation is provided in Appendix A of this article (see also [54]).

### D. Indistinguishable holes (partons)

To use the effective parton theory for describing doped holes in the $t - J$ or $t - J_z$ model, we next consider the case of indistinguishable partons. I.e. the string states must be invariant, up to an overall sign, if holes $n = 1$ and $n = 2$ are exchanged. To treat both partons on equal footing, we need to go beyond the asymmetric approximation made earlier, where rotational basis states were defined around parton $n = 1$.

The required (anti)-symmetrization procedure further complicates the use of the truncated basis. In this section we will show that a suitable generalization of the truncated basis allows to keep the rotational quantum numbers, and derive effective Hamiltonians separately for fermions and bosons. As a result, $m_4$ remains a good quantum number at C4IM.

Together, the obtained fermionic and bosonic eigenstates span the space of distinguishable partons. When $t_1 = t_2$, any bosonic (fermionic) eigenstate we identify also constitutes an eigenstate of distinguishable partons – however, the use of the (anti-) symmetrized truncated basis leads to an improved variational energy going beyond the asymmetric approximation from Sec. III C; there, the rotational basis was defined around just one of the two partons, breaking the symmetry implied by $t_1 = t_2$.

The results of our approach will be presented in the subsequent section IV, which can be understood without following the technical details derived in the remainder of this section. However, subsection III D 5 may be of interest, where we discuss the relation between effective bosonic / fermionic string states and the microscopic AFM $t - J_{(z)}$ models composed of bosons or fermions, respectively.

### 1. First quantization formalism

We start by defining the permutation operator which exchanges the labels of the holes,

$$\hat{\mathcal{P}}|\boldsymbol{x}_1, \Sigma\rangle = |\boldsymbol{x}_2, \overline{\Sigma}\rangle, \quad (33)$$

where $\overline{\Sigma}$ describes the same string as $\Sigma$ but starting from the opposite end.

If we consider the case $t_1 = t_2 = t$ in our model, the Hamiltonian $\hat{\mathcal{H}}^{\text{eff}}$ in Eq. (10) commutes with the permutation operator, $[\hat{\mathcal{H}}^{\text{eff}}, \hat{\mathcal{P}}] = 0$. Hence every eigenstate belongs to one of two classes, either fermionic (with $\hat{\mathcal{P}}$-eigenvalue $-1$) or bosonic (with $\hat{\mathcal{P}}$-eigenvalue $+1$).

Making use of this symmetry is still challenging, however, due to the exponential size of the string Hilbert space. Moreover, the string states $|\boldsymbol{k}, \ell_\Sigma, \boldsymbol{m}\rangle$ introduced above in general do not have well-defined exchange statistics, i.e. they are not eigenstates of $\hat{\mathcal{P}}$. We will discuss below how this problem can be avoided and proper string-states can be defined.

*a. String-length $\ell_\Sigma = 1$ states.–* The high-symmetry states at the shortest string length $\ell_\Sigma = 1$ are of particular importance for defining quasiparticle weights of pairs [39] later on. At the C4IM $\boldsymbol{k}^{\text{C4IM}}$ (for the considered square lattice these are $\boldsymbol{k}^{\text{C4IM}} = \boldsymbol{0}$ and $\boldsymbol{k}^{\text{C4IM}} = (\pi, \pi)$) one obtains

$$\hat{\mathcal{P}}\,|\boldsymbol{k}^{\text{C4IM}}, \ell_\Sigma = 1, m_4\rangle = (-1)^{m_4}$$
$$\times\, e^{i(\boldsymbol{e}_x + \boldsymbol{e}_y)\cdot\boldsymbol{k}^{\text{C4IM}}/2}\,|\boldsymbol{k}^{\text{C4IM}}, \ell_\Sigma = 1, m_4\rangle, \quad (34)$$

i.e. these states are purely fermionic / bosonic respectively. Concretely, at $\boldsymbol{k}^{\text{C4IM}} = \boldsymbol{0}$ the states $m_4 = 0, 2$ ($m_4 = 1, 3$) are bosonic (fermionic) respectively. Vice-versa, at $\boldsymbol{k}^{\text{C4IM}} = (\pi, \pi)$ the states $m_4 = 0, 2$ ($m_4 = 1, 3$) are fermionic (bosonic).

It may seem counter-intuitive at first to have (spin-less) fermionic pairs of holes with $s$-wave pairing symmetry at $\boldsymbol{k} = (\pi, \pi)$. However, the center of mass momentum $\boldsymbol{k} = (\pi, \pi)$ effectively leads to an anti-symmetric wavefunction under exchange of the $A$ and $B$ sublattices. This ensures correct overall fermionic statistics even in the presence of an $s$-wave pairing symmetry, in the effective string model. We will discuss in Sec. IV A 2 how this relates to pairs in the microscopic Néel state, which breaks the discrete translational symmetry of the square lattice.

*b. General string states.–* For general string states, some additional insights can be gained at C4IM. For example, we find that the rotational ground states with $\boldsymbol{m} = \boldsymbol{0}$ satisfy: (i) all $\boldsymbol{k} = \boldsymbol{0}$ states are bosonic,

$$\hat{\mathcal{P}}|\boldsymbol{k} = \boldsymbol{0}, \ell_\Sigma, \boldsymbol{m} = \boldsymbol{0}\rangle = |\boldsymbol{k} = \boldsymbol{0}, \ell_\Sigma, \boldsymbol{m} = \boldsymbol{0}\rangle; \quad (35)$$

and, (ii), the statistics of states at $\boldsymbol{k} = \boldsymbol{\pi} = (\pi, \pi)$ alternates with the string length,

$$\hat{\mathcal{P}}|\boldsymbol{k} = \boldsymbol{\pi}, \ell_\Sigma, \boldsymbol{m} = \boldsymbol{0}\rangle = (-1)^{\ell_\Sigma}|\boldsymbol{k} = \boldsymbol{\pi}, \ell_\Sigma, \boldsymbol{m} = \boldsymbol{0}\rangle. \quad (36)$$

### 2. (Anti-) symmetrized truncated basis

As explained above, the rotational basis states generally do *not* have proper exchange statistics. To enforce the latter, we can explicitly (anti-) symmetrize the basis states by defining new basis states:

$$|\boldsymbol{k}, \ell_\Sigma, \boldsymbol{m}, \mu\rangle = (1 + \mu\hat{\mathcal{P}})\,|\boldsymbol{k}, \ell_\Sigma, \boldsymbol{m}\rangle. \quad (37)$$

Here $\mu = \pm 1$ denotes bosonic (fermionic) states when $\mu = +1$ ($\mu = -1$, respectively). By restricting states in Eq. (37) to the lowest rotational quantum numbers $\boldsymbol{m}$, a new truncated basis with well-defined exchange statistics is obtained. However, the new states are no longer orthonormal, and we use a Gram-Schmidt procedure to construct an orthonormal basis (ONB) in the linear space spanned by states in Eq. (37).

*a. Gram-Schmidt procedure.–* For each given set of good quantum numbers $\boldsymbol{k}$, $\ell_\Sigma$ and $\mu$ we use the standard Gram-Schmidt method to construct ONB states by mixing different rotational states $\boldsymbol{m}$. Denoting the new ONB states by $\tilde{\boldsymbol{m}}$, we have the general representation

$$|\boldsymbol{k}, \ell_\Sigma, \tilde{\boldsymbol{m}}, \mu\rangle = \sum_{\boldsymbol{m}} \tilde{C}_{\boldsymbol{m}, \tilde{\boldsymbol{m}}}(\boldsymbol{k}, \ell_\Sigma, \mu)\,|\boldsymbol{k}, \ell_\Sigma, \boldsymbol{m}, \mu\rangle \quad (38)$$

with coefficients determined by the matrix $\tilde{C}(\boldsymbol{k}, \ell_\Sigma, \mu)$. Specifically, we choose

$$|\tilde{\boldsymbol{m}} = \boldsymbol{0}\rangle = \mathcal{N}_0|\boldsymbol{m} = \boldsymbol{0}\rangle,$$
$$\dots$$
$$|\tilde{\boldsymbol{m}}\rangle = \mathcal{N}_{\tilde{\boldsymbol{m}}}\left(|\boldsymbol{m}\rangle - \sum_{\tilde{\boldsymbol{M}} < \tilde{\boldsymbol{m}}} |\tilde{\boldsymbol{M}}\rangle\langle\tilde{\boldsymbol{M}}|\boldsymbol{m}\rangle\right),$$

where we dropped all labels $\boldsymbol{k}$ and $\ell_\Sigma$ for simplicity and $\mathcal{N}_{\tilde{\boldsymbol{m}}}$ denotes normalization constants.

*b. Calculating overlaps of string states.–* To calculate overlaps between the old and new basis states, we use the following explicit expression for overlaps between the symmetrized states defined in Eq. (37):

$$\langle\ell, m_4', m_3'|\,\ell, m_4, m_3\rangle = 2\delta_{m_4', m_4}\delta_{m_3', m_3} + \frac{\mu}{4 \times 3^{\ell-1}}$$
$$\times \sum_{\nu, \nu'=0}^{4} e^{i\frac{\pi}{2}(\nu m_4 - \nu' m_4')} \sum_{\lambda, \lambda'=1}^{3} e^{i\frac{2\pi}{3}(\lambda m_3 - \lambda' m_3')}$$
$$\times\left(\sum_{\substack{\Sigma:\ \ell_\Sigma = \ell \\ \varphi_\Sigma = \frac{\pi}{2}\nu',\ \vartheta_\Sigma = \frac{\pi}{2}\lambda' \\ \overline{\varphi}_\Sigma = \frac{\pi}{2}\nu,\ \overline{\vartheta}_\Sigma = \frac{\pi}{2}\lambda}} e^{i\boldsymbol{k}\cdot\boldsymbol{R}_\Sigma} + \sum_{\substack{\Sigma:\ \ell_\Sigma = \ell \\ \varphi_\Sigma = \frac{\pi}{2}\nu,\ \vartheta_\Sigma = \frac{\pi}{2}\lambda \\ \overline{\varphi}_\Sigma = \frac{\pi}{2}\nu',\ \overline{\vartheta}_\Sigma = \frac{\pi}{2}\lambda'}} e^{-i\boldsymbol{k}\cdot\boldsymbol{R}_\Sigma}\right)$$
$$(39)$$

again dropping $\boldsymbol{k}$ and $\mu$ labels for simplicity; here we only included $m_4$ and $m_3^{(1)}$ rotational states. In the last line we sum over all string configurations $\Sigma$ whose length $\ell_\Sigma = \ell$ equals the length $\ell$ in the considered string sector, and with constraints on the last string segments (see Fig. 5):

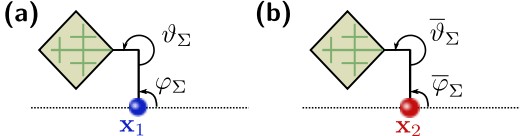

FIG. 5. The orientations of the first two string segments start-
ing from hole 1 in (a) (hole 2 in (b)) are labeled by the angle
$\varphi_\Sigma$ ($\overline{\varphi}_\Sigma$) relative to the x-axis, and the angle $\vartheta_\Sigma$ ($\overline{\vartheta}_\Sigma$) mea-
sured relative to the preceding string segment.

$\varphi_\Sigma$ denotes the angle of the first string segment starting
at the central hole ($n = 1$) measured relative to the x-
axis; $\vartheta_\Sigma$ denotes the angle of the second string segment
starting at the central hole ($n = 1$) measured relative to
first string segment; $\boldsymbol{R}_\Sigma = \boldsymbol{x}_2 - \boldsymbol{x}_1$ denotes the vector in
the 2D square lattice connecting the first hole at $\boldsymbol{x}_1$ to
the second hole at $\boldsymbol{x}_2$; the angles $\overline{\varphi}_\Sigma$ and $\overline{\vartheta}_\Sigma$ are defined
like $\varphi_\Sigma$ and $\vartheta_\Sigma$ but starting from the other hole ($n = 2$),
as shown in Fig. 5.

For maximum string lengths up to around $\ell \simeq 13$ or
slightly larger, the overlaps in Eq. (39) can be calculated
by an exact summation over all string states. For larger
maximum string lengths, Metropolis Monte-Carlo sam-
pling over string configurations can be used to obtain
good estimates for the overlaps.

### 3. Effective Hamiltonian

To obtain the Hamiltonian in the Gram-Schmidt ONB
basis $|\boldsymbol{k}, \ell_\Sigma, \tilde{\boldsymbol{m}}\rangle$, we project the effective Hamiltonian to
the symmetrized subspace,

$$\tilde{\mathcal{H}} = \tilde{P} \ \hat{\mathcal{H}}^{\text{eff}} \ \tilde{P} \tag{40}$$

where $\tilde{P} = \sum_{\ell_\Sigma} \sum_{\tilde{\boldsymbol{m}}} |\ell_\Sigma, \tilde{\boldsymbol{m}}\rangle\langle\ell_\Sigma, \tilde{\boldsymbol{m}}|$, and directly calcu-
late its matrix elements $\langle\ell'_\Sigma, \tilde{\boldsymbol{m}}'|\tilde{\mathcal{H}}|\ell_\Sigma, \tilde{\boldsymbol{m}}\rangle$.

To this end, we first use Eq. (38) to express the ONB
vectors in terms of the non-ONB states. The action of
the Hamiltonian on the latter, $\hat{\mathcal{H}}^{\text{eff}}|\boldsymbol{k}, \ell, \boldsymbol{m}, \mu\rangle$, is known
from the case of distinguishable holes, noting that the
(anti-) symmetrization $(1 + \mu\hat{\mathcal{P}})$ is a linear operation.
This leads to the processes $\propto J_\pm^{(n)}(\boldsymbol{k}; \boldsymbol{m}', \boldsymbol{m})$ and $V(\ell_\Sigma)$.

Thus we obtain an expression for $\tilde{\mathcal{H}}|\boldsymbol{k}, \ell, \tilde{\boldsymbol{m}}, \mu\rangle$ as a sum
over the non-ONB basis states $|\boldsymbol{k}, \ell, \boldsymbol{m}, \mu\rangle$ weighted by
coefficients $\tilde{C}$ and couplings $J_\pm^{(n)}$ or $V(\ell_\Sigma)$. This allows
us to express the non-ONB states in terms of ONB states
again by

$$\tilde{P}|\boldsymbol{k}, \ell_\Sigma, \boldsymbol{m}, \mu\rangle = \sum_{\tilde{\boldsymbol{m}}} C_{\tilde{\boldsymbol{m}}, \boldsymbol{m}}(\boldsymbol{k}, \ell_\Sigma, \mu) \ |\boldsymbol{k}, \ell_\Sigma, \tilde{\boldsymbol{m}}, \mu\rangle, \tag{41}$$

with overlaps

$$C_{\tilde{\boldsymbol{m}}, \boldsymbol{m}}(\boldsymbol{k}, \ell_\Sigma, \mu) = \langle\boldsymbol{k}, \ell_\Sigma, \tilde{\boldsymbol{m}}| \ \boldsymbol{k}, \ell_\Sigma, \boldsymbol{m}\rangle. \tag{42}$$

Finally, this leads to the effective Hamiltonian in the
ONB basis. It is most conveniently expressed by start-
ing from the distinguishable-hole Hamiltonian in square

matrix form:

$$\underline{H} = \underline{V} + \left(\underline{\underline{J_-}} + \underline{\underline{J_+}}\right), \tag{43}$$

where $\underline{V}$ is a diagonal matrix with entries $V(\ell_\Sigma) \otimes 1_{\boldsymbol{m}}$.
Moreover, $\underline{\underline{J_\pm}}$ are upper (lower) block-band matrices con-
necting states at $\ell_\Sigma$ and $\ell_\Sigma - 1$ ($\ell_\Sigma + 1$) respectively,
and mixing $\boldsymbol{m}$ quantum numbers in the truncated basis;
i.e. the matrix elements of $\underline{\underline{J_\pm}}$ are given by $J_\pm^{(1)} + J_\pm^{(2)}$.
Defining block-diagonal matrices $\underline{C}$ and $\underline{\tilde{C}}$ with rectan-
gular blocks $C_{\tilde{\boldsymbol{m}}, \boldsymbol{m}}(\ell_\Sigma)$ and $\tilde{C}_{\boldsymbol{m}, \tilde{\boldsymbol{m}}}(\ell_\Sigma)$ at each block of
states with string length $\ell_\Sigma$, we obtain for indistinguish-
able holes in the ONB basis:

$$\underline{\tilde{H}} = \underline{C} \ \underline{H} \ \underline{\tilde{C}}. \tag{44}$$

Note that through the matrices $\underline{C}$ and $\underline{\tilde{C}}$ the Hamilto-
nian $\underline{\tilde{H}}$ depends on the particle statistics $\mu$; i.e. different
Hamiltonians are obtained for bosonic, fermionic and dis-
tinguishable holes.

### 4. Quasiparticle weights

For the eigenstates $|\psi_n(\boldsymbol{k}, \mu)\rangle$ of the Hamiltonian (44)
in the ONB basis, we define the corresponding quasi-
particle weights by their overlaps squared with (anti-)
symmetrized states at string length $\ell_\Sigma = 1$:

$$Z_n(\boldsymbol{k}, m_4, \mu) = |\langle\psi_n(\boldsymbol{k}, \mu)| \ \boldsymbol{k}, 1, m_4, \mu\rangle|^2. \tag{45}$$

This result can be expressed conveniently in terms of the
matrices $\tilde{C}_{\boldsymbol{m}, \tilde{\boldsymbol{m}}}$ and $C_{\tilde{\boldsymbol{m}}, \boldsymbol{m}}$ and projectors $\underline{\underline{P_{\ell=1}^{m_4}}}$ to the
string-length $\ell = 1$ states with rotational quantum num-
ber $m_4$. We obtain:

$$Z_n(\boldsymbol{k}, m_4, \mu) = \langle\psi_n(\boldsymbol{k}, \mu)|\underline{C} \ \underline{\underline{P_{\ell=1}^{m_4}}} \ \underline{C}^\dagger|\psi_n(\boldsymbol{k}, \mu)\rangle. \tag{46}$$

### 5. Exchange statistics in microscopic and effective models

So far, our discussion of exchange statistics was only
on the level of the effective string model, where parti-
cle exchange is defined through the application on the
permutation operator $\hat{\mathcal{P}}$, see Eq. (33). On the other
hand, the effective string states $|\boldsymbol{x}, \Sigma\rangle$ can be directly
related to two-hole states $|\Psi(\boldsymbol{x}, \Sigma)\rangle$ in a classical Néel
background composed of constituents $\hat{c}_{j,\sigma}$, see Secs. II
and III A. To this end, one starts from the classical Néel
state $|\text{N}\rangle$. Next, the first hole is created at site $\boldsymbol{x}$ and the
second hole is created next to $\boldsymbol{x}$ at $\boldsymbol{R}_2 = \boldsymbol{x} + \boldsymbol{\delta}_1$ along
the direction of the first string segment in $\Sigma$, by applying
$\hat{c}_{\boldsymbol{x}+\boldsymbol{\delta}_1,\sigma_2}\hat{c}_{\boldsymbol{x},\sigma_1}|\text{N}\rangle$. Then the second hole is moved around,
along the directions $\boldsymbol{\delta}_n$ of subsequent string segments in

$\Sigma$, by applying hopping terms $\sum_\sigma \hat{c}^\dagger_{\boldsymbol{R}_n+\boldsymbol{\delta}_n,\sigma}\hat{c}_{\boldsymbol{R}_n}$, where $\boldsymbol{R}_{n+1} = \boldsymbol{R}_n + \delta_n$.

To understand how the exchange statistics $\mu$ in the effective string model relates to the statistics of the microscopic constituents $\hat{c}_{\boldsymbol{j},\sigma}$ of the Néel AFM, we start from a parton construction representing the underlying spins as $\hat{c}_{\boldsymbol{j},\sigma} = \hat{h}^\dagger_{\boldsymbol{j}}\hat{f}_{\boldsymbol{j},\sigma}$, subject to the local constraints $\sum_\sigma \hat{f}^\dagger_{\boldsymbol{j},\sigma}\hat{f}_{\boldsymbol{j},\sigma} + \hat{h}^\dagger_{\boldsymbol{j}}\hat{h}_{\boldsymbol{j}} = 1$. One can choose different combinations of parton statistics $\mu_{f,h} = \pm 1$, as long as $\mu_c = \mu_f\mu_h$, where $\mu_c = +1$ ($\mu_c = -1$) if $\hat{c}_{\boldsymbol{j},\sigma}$ are bosons (fermions).

The simplest way to map effective string states $|\boldsymbol{x},\Sigma\rangle$ to microscopic two-hole states $|\Psi(\boldsymbol{x},\Sigma)\rangle$ is to choose bosonic spinons $\mu_f = +1$, i.e. $\hat{f}_{\boldsymbol{j},\sigma}$ are Schwinger bosons, and $\mu_h = \mu_c$. The spinons $\hat{f}_{\boldsymbol{j},\sigma}$ keep track of how the spin pattern is distorted by the hole motion creating the string $\Sigma$, and since we chose bosonic spinons the order of the spins in the background is irrelevant. Exchanging the two chargons $\hat{h}_{\boldsymbol{x}_1}\hat{h}_{\boldsymbol{x}_2} = \mu_h\hat{h}_{\boldsymbol{x}_2}\hat{h}_{\boldsymbol{x}_1}$ keeps track of additional minus signs in case $\mu_h = -1$. To reflect the resulting exchange statistics $\mu_h = \mu_c$ correctly, the statistics of the effective string states should equal $\mu = \mu_c$.

Making the opposite choice $\mu_f = -1$ and $\mu_h = -\mu_c$, i.e. choosing fermionic spinons $\hat{f}_{\boldsymbol{j},\sigma}$, leads to the same result, $\mu = \mu_c$. However, in this case one has to keep track of exchange signs $\mu_f$ picked up when spin operators in the background are exchanged, which requires additional book keeping.

## IV. RESULTS

We compare the predictions by our string-based model of fermionic hole pairing to fully numerically obtained two-hole spectra (from matrix product states) on four-leg cylinders in Ref. [39]. There we find good agreement for the $t-J_z$ and $t-J$ models, and we discuss how accurately our string model is able to describe the numerical observations. The main focus in the present article is on understanding the predictions of our effective model, the accuracy of the truncated bases we use, and the analytical insights that can be gained from our calculations.

### A. Two-hole spectra

We start by calculating the two-hole eigenstates along a high-symmetry cut through the Brillouin zone in Fig. 6. We compare our results for different statistics of the holes: fermionic (relevant to pairing in the doped Hubbard model) and bosonic (as a theoretical reference); the case of distinguishable dopants (relevant to Hubbard-Mott excitons [35]) corresponds to the combined bosonic and fermionic eigenstates. As a theoretical reference, we also show results from the less accurate asymmetric approximation, i.e. without (anti-) symmetrizing rotational

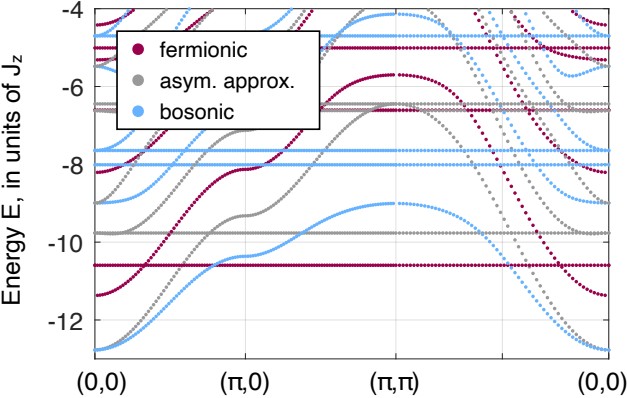

FIG. 6. Two-hole eigenstates from the truncated string basis calculation. We compare fermionic (red) and bosonic (blue) holes, and show results from the asymmetric approximation (gray). The truncated basis used for the calculations includes all $m_4 = 0,..,3$ and $m_3^{(1)} = 0,1,2$ sectors, and string lengths up to $\ell_{\max} = 11$. We considered $t/J_z = 3$ and a string potential for an Ising background.

states in the truncated basis. Other parameters are identical, we assume $t/J_z = 3$ and calculate the string potential Eq. (13) for a $t-J_z$ model i.e. using Eq. (14) for all cases.

We observe several striking features, discussed in more detail shortly: (i) in all cases, the lowest energy state has zero momentum $\boldsymbol{k} = \boldsymbol{0}$; (ii) the ground state of distinguishable holes is bosonic and is captured by the asymmetric approximation; (iii) its energy is significantly below the lowest-energy fermionic state; (iv) the lowest-energy states are highly dispersive, with an effective mass scaling as $M_{\mathrm{hh}} \propto 1/t$; (v) in addition to several strongly dispersing bands, we observe numerous exactly flat bands; (vi) while their energy differs between different hole statistics, they are always present; (vii) in the fermionic case, the flat band constitutes the lowest-energy state over a significant portion of the Brillouin zone, except around $\boldsymbol{k} = \boldsymbol{0}$.

Some comments are in order: Regarding (i), we note that for the fermionic case, the dispersive $\boldsymbol{k} = 0$ band is only slight lower in energy than the lowest flat band. This competition is found to be even more pronounced for smaller values of $t/J_z$ (not shown). For $t \ll J_z$ we find that the dispersive state becomes degenerate with the flat-band state at $\boldsymbol{k} = 0$.

Regarding (ii), note that the lower variational energy at $\boldsymbol{k} \neq 0$ of the bosonic ground state demonstrates higher accuracy of the (anti-) symmetrized states as compared to the asymmetric approximation.

Regarding (iii), it has been pointed out previously by Trugman [13] that the fermionic sign effectively frustrates the hopping Hamiltonian of two fermionic holes connected by a string. We believe this explains the increase of their energy relative to the bosonic or distinguishable holes, as observed in Fig. 6.

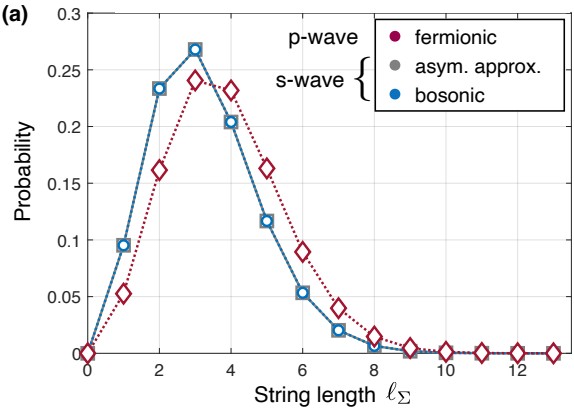 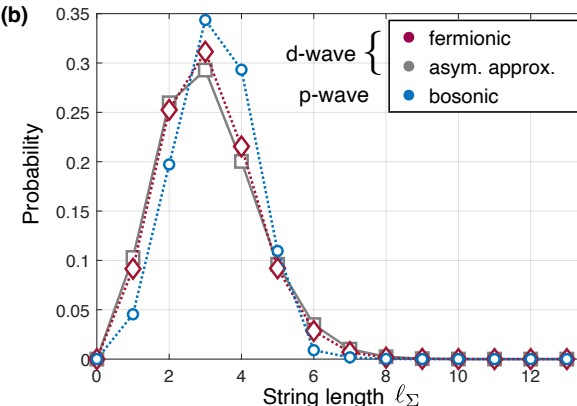

FIG. 7. String-length distributions for fermionic (red) and bosonic (blue) holes in (a) the ground state around $\boldsymbol{k} = \boldsymbol{0}$ and (b) in the lowest-energy state around $\boldsymbol{k} = (\pi, \pi)$. Predictions by the asymmetric approximation are also shown (gray). The rotational quantum numbers of all states can be extracted by analyzing spectral weights, and our results are indicated in the legend: s-wave ($m_4 = 0$), p-wave ($m_4 = 1$), d-wave ($m_4 = 2$). Note in (a) the bosonic prediction coincides with the asymmetric approximation. Throughout we used the truncated basis including all $m_4 = 0, .., 3$ and $m_3^{(1)} = 0, 1, 2$ sectors, and string lengths up to $\ell_{\max} = 13$. We considered $t/J_z = 3$ and a string potential for an Ising background.

Regarding (iv), it has long been understood that the ability of one hole to follow the other may lead to a highly mobile bound state with bandwidth $\propto t$ in the $t - J_z$ model. An estimate for the effective mass, $M_{\mathrm{hh}}^{-1} = t\sqrt{3}$, was derived for $t \gg J_z$ from a string model in Ref. [42]. The inclusion of quantum fluctuations in the $t - J$ model is expected to cause additional polaronic dressing of these states and a corresponding mass enhancement.

Regarding (v), the existence of flat bands corresponding to $M_{\mathrm{hh}} \to \infty$ in an effective string model has been predicted by Shraiman and Siggia [14], although they used a slightly different procedure to truncate their string basis. This suggests that flat bands of hole pairs are not an artifact but robust excitations of the system. Indeed, in our recent numerical work [39] we found strong evidence for the existence of flat bands of hole pairs in the $t - J_z$ model.

As in many flat-band systems, we believe that destructive quantum interference between different paths underlies the formation of self-localized flat-band states. Notably, we checked that this does not limit the string length of the pair: As shown in Fig. 7, the string-length distribution of the dispersive bound state around $\boldsymbol{k} = 0$ (a) is qualitatively similar – with a broad peak around $\ell_{\Sigma} = 3$ – to the flat-band bound state around $\boldsymbol{k} = (\pi, \pi)$. While different statistics of the holes and $m_4$ quantum numbers usually correspond to small differences in the string-length histograms, their overall shape is always similar with an average string-length $\langle \ell_{\Sigma} \rangle \approx 3 - 4$ for $t/J_z = 3$.

Regarding (vii), we note that additional polaronic dressing by spin-waves [30] or phonons in a solid is expected to lower the energy of the flat-band state further compared to the dispersive band, since the large recoil energy $\propto t$ associated with the dispersive band suppresses polaronic dressing. Moreover, strong interactions

between the pairs may favor occupying the self-localized flat-band states, where localization costs no kinetic energy while the interaction energy can be minimized.

### 1. Spectral weights

So far we have only calculated the energies of string-paired eigenstates in Fig. 6. To reveal the nature of different states, we calculate their spectral weights $Z_n(\boldsymbol{k}, m_4)$ for different hole statistics, i.e. their overlaps squared with string-length $\ell_{\Sigma} = 1$ states of the same symmetries, see Eq. (45). For presentation purposes, the obtained spectral lines are broadened with a Gaussian of width $\sigma = J_z/5$, and the integrated spectral weight per peak equals the corresponding quasiparticle weight. Our results are shown as momentum cuts for different values of $m_4$ in Figs. 8-10.

Before we discuss our results, we emphasize that while the overlaps with string-length one states of a given momentum $\boldsymbol{k}$ and rotational quantum number $m_4$ are well-defined, the paired eigenstates only have a well-defined rotational eigenvalue for C4IM. For general $\boldsymbol{k} \notin$ C4IM, the momentum $\boldsymbol{k}$ explicitly breaks the $C_4$ symmetry and different rotational states can hybridize. Moreover, the A- and B- sublattice degree of freedom on the square lattice allows to realize $s/ d$-wave states of indistinguishable fermions ($p / f$-wave states of indistinguishable bosons) at general $\boldsymbol{k}$, whose relation to microscopic states in a Néel AFM we discuss further in subsection IV A 2.

In Fig. 8 we show our results for the case of distinguishable holes, relevant e.g. to models of Mott excitons [35] or mixD bilayer pairing [42]. In the $s$-wave channel (a) we observe only dispersive bands and a collapse of spectral weight at high energies around $\boldsymbol{k} = (\pi, \pi)$. The latter effect was recently found numerically in a

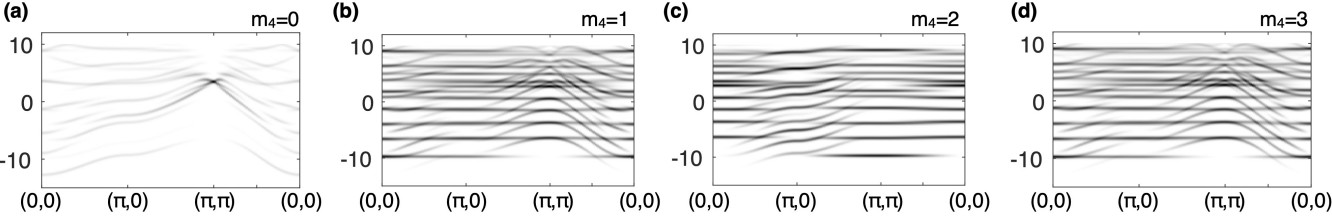

FIG. 8. Two-hole spectra for distinguishable holes connected by a string, calculated using the asymmetric approximation, along a high-symmetry cut through the Brillouin zone of the square lattice. The plots are obtained from a spectral decomposition and using our semi-analytical theory with a truncated basis including all $m_4$ and $m_3^{(1)}$ sectors, up to a maximum string length $\ell_{max} = 11$. We considered $t/J_z = 3$ and a string potential for an Ising background.

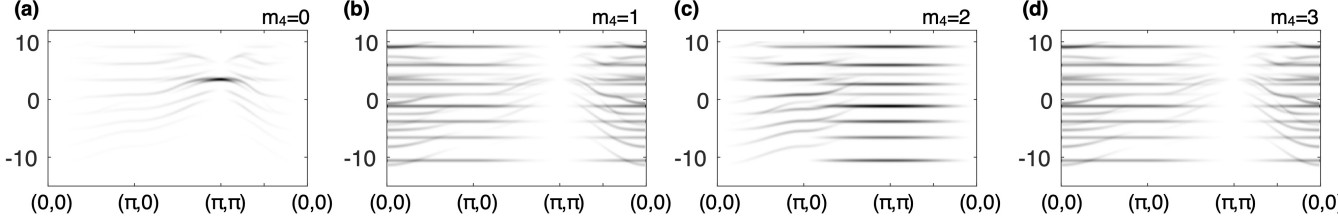

FIG. 9. Two-hole spectra as in Fig. 8, but for fermionic holes. Again, all $m_4$ and $m_3^{(1)}$ sectors were included and a maximum string length $\ell_{max} = 11$ was used. We considered $t/J_z = 3$ and a string potential for an Ising background.

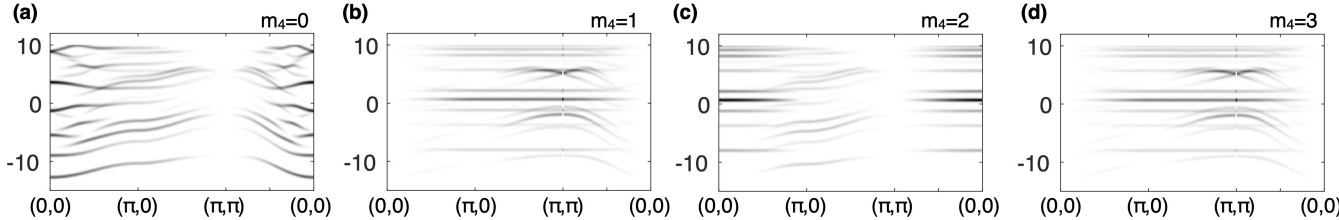

FIG. 10. Two-hole spectra as in Figs. 8, 9, but for bosonic holes. Again, all $m_4$ and $m_3^{(1)}$ sectors were included and a maximum string length $\ell_{max} = 11$ was used. We considered $t/J_z = 3$ and a string potential for an Ising background.

microscopic bilayer model [42]. In the other channels $m_4 = 1, 2, 3$ shown in (b)-(d) we observe a mix of many flat and highly dispersive bands, where $d$-wave states are flat around $\boldsymbol{k} = (\pi, \pi)$ and dispersive around the corners of the Brillouin zone $\boldsymbol{k} = (0, \pi)$.

In Fig. 9 we show the same plot for fermionic holes. A first striking difference is the complete suppression of spectral weight around $\boldsymbol{k} = 0$ (for $s$- and $d$-wave pairs) and around $\boldsymbol{k} = (\pi, \pi)$ (for $p$- and $f$-wave pairs). This follows directly from the fixed and alternating statistics of the string-length $\ell_\Sigma = 1$ states at those momenta, derived in Eq. (34). Moreover, we find in (a) that the collapse of spectral weight at high energies in the $s$-wave channel around $\boldsymbol{k} = 0$ remains a robust feature for fermionic holes. This has been observed in earlier exact diagonalization calculations for the $t - J$ model [55] and confirmed by our recent numerical DMRG study [39] for the $t - J$ and $t - J_z$ models.

At low energies in Fig. 9 we only find spectral weight in the $p$-, $d$- and $f$-wave channels, corresponding to the lowest-energy flat band. As we discuss further below, the

prediction of a flat band with $d$-wave character around $\boldsymbol{k} = (\pi, \pi)$ is consistent with earlier exact diagonalization results reporting narrow $d$-wave and $p$-wave peaks at low energies [55], and further corroborated in [39]. In addition we predict a strongly dispersive band at low energies around $\boldsymbol{k} = 0$ which contributes only little spectral weight, however.

In Fig. 10 we show spectra for bosonic holes. As in the fermionic case, some regions have zero spectral weight owing to the nature of string-length one states, see Eq. (34). In the bosonic case, as a consequence, no collapse of spectral weight at high energies can be observed. At low energies, the spectral weight is dominated by the lowest dispersive band. At higher energies, flat bands with $p$-, $d$- and $f$-wave character can be observed.

The spectrum for distinguishable holes, but with $t_1 = t_2 = t$ equal, can be obtained by the sum of the fermionic and bosonic spectrum. Qualitatively, this procedure matches our results in Fig. 8; but note that the data in Fig. 8 is based on the less accurate asymmetric approximation, which leads to some quantitative deviations.

### 2. Relation to pairs in a Néel-ordered state

To understand the relation of the two-hole spectra calculated in Figs. 8 - 10 to studies of pairs in the $t - J_{(z)}$ or Hubbard models, see e.g. [55], note that we have so far worked in an effective parton basis. Namely, the spin background was assumed to define a featureless vacuum state and our starting point was the two-hole string basis $|\boldsymbol{x}_1, \Sigma\rangle$ defined in Eq. (4). Since a proper two-hole spectral function should connect the undoped state (our vacuum) to the paired eigenstates, the structure of the former matters.

To understand the effect of the spin background, we consider starting from a classical Néel state $|\mathrm{N}\rangle$, i.e. one of the symmetry-broken ground states of the 2D Ising model. We will assume that $\uparrow$ ($\downarrow$) spins occupy the A (B) sublattice and have bosonic or fermionic statistics characterized by $\mu$. This state breaks the lattice-translational symmetry, which leads to momenta well-defined only within the 2-site magnetic Brillouin zone (MBZ). Nevertheless, within the effective string model with its one-site unit-cell, Umklapp scattering from outside to within the MBZ is not possible. Hence the two-hole bandstructure within the MBZ is obtained by simply folding the full dispersion into the MBZ.

In this process the rotational quantum numbers remain unchanged. As a result we can identify the following string-model states $|\boldsymbol{k}, m_4\rangle$ for any statistics $\mu$

$$|\boldsymbol{k} = (\pi, \pi), m_4\rangle \equiv 2|\boldsymbol{k} = 0, m_4\rangle_{\mathrm{N}} \qquad (47)$$

with states $|\boldsymbol{k}, m_4\rangle_{\mathrm{N}}$ in the Néel background (proof below). I.e. for fermions, where only $m_4 = 0, 2$ states exist at $\boldsymbol{k} = (\pi, \pi)$, we obtain the corresponding $s$- and $d$-wave peaks around $\boldsymbol{k} = 0$ in the MBZ; for bosons, where only $m_4 = 1, 3$ states exist at $\boldsymbol{k} = (\pi, \pi)$, we obtain the corresponding $p$- and $f$-wave peaks around $\boldsymbol{k} = 0$ in the MBZ.

To show Eq. (47), we note that the two-hole string state corresponds to

$$|\boldsymbol{k}, m_4\rangle = \frac{1}{\sqrt{V}} \sum_{\boldsymbol{j}} e^{i\boldsymbol{k} \cdot \boldsymbol{j}} \hat{h}_{\boldsymbol{j}}^\dagger \sum_{\boldsymbol{i}:\langle \boldsymbol{i},\boldsymbol{j}\rangle} e^{i\frac{\pi}{2} m_4 \nu_{\langle \boldsymbol{i},\boldsymbol{j}\rangle}} \hat{h}_{\boldsymbol{i}}^\dagger |\mathrm{N}\rangle, \quad (48)$$

where $\frac{\pi}{2}\nu_{\langle \boldsymbol{i},\boldsymbol{j}\rangle} = \arg(\boldsymbol{i} - \boldsymbol{j})$ is the angle of $\boldsymbol{i}$ relative to $\boldsymbol{j}$. The hole creation operator acts on the Néel state as

$$\hat{h}_{\boldsymbol{j}}^\dagger|\mathrm{N}\rangle = \hat{c}_{\boldsymbol{j},\uparrow}|\mathrm{N}\rangle \qquad \boldsymbol{j} \in \mathrm{A},$$
$$\hat{h}_{\boldsymbol{j}}^\dagger|\mathrm{N}\rangle = \hat{c}_{\boldsymbol{j},\downarrow}|\mathrm{N}\rangle \qquad \boldsymbol{j} \in \mathrm{B},$$

and the state $|\boldsymbol{k}, m_4\rangle$ becomes

$$|\boldsymbol{k}, m_4\rangle = \frac{1}{\sqrt{V}} \sum_{\boldsymbol{j} \in \mathrm{A}} e^{i\boldsymbol{k} \cdot \boldsymbol{j}} \hat{c}_{\boldsymbol{j},\uparrow} \sum_{\boldsymbol{i}:\langle \boldsymbol{i},\boldsymbol{j}\rangle} e^{i\frac{\pi}{2} m_4 \nu_{\langle \boldsymbol{i},\boldsymbol{j}\rangle}} \hat{c}_{\boldsymbol{i},\downarrow}|\mathrm{N}\rangle +$$
$$+ \frac{\mu}{\sqrt{V}} \sum_{\boldsymbol{i} \in \mathrm{A}} \hat{c}_{\boldsymbol{i},\uparrow} \sum_{\boldsymbol{j}:\langle \boldsymbol{i},\boldsymbol{j}\rangle} e^{i\frac{\pi}{2} m_4(\nu_{\langle \boldsymbol{j},\boldsymbol{i}\rangle}+2)} \hat{c}_{\boldsymbol{j},\downarrow} e^{i\boldsymbol{k} \cdot \boldsymbol{j}}|\mathrm{N}\rangle. \quad (49)$$

In the second line we exchanged the order of the operators, which yields the exchange sign $\mu$, and used that $\nu_{\langle \boldsymbol{i},\boldsymbol{j}\rangle} = \nu_{\langle \boldsymbol{j},\boldsymbol{i}\rangle} + 2 \mod 4$. At $\boldsymbol{k} = (\pi, \pi) \equiv \boldsymbol{\pi}$ we obtain

$$|\boldsymbol{\pi}, m_4\rangle = \frac{1 - \mu e^{i\pi m_4}}{\sqrt{V}} \sum_{\boldsymbol{j}} \hat{c}_{\boldsymbol{j},\uparrow} \sum_{\boldsymbol{i}:\langle \boldsymbol{i},\boldsymbol{j}\rangle} e^{i\frac{\pi}{2} m_4 \nu_{\langle \boldsymbol{i},\boldsymbol{j}\rangle}} \hat{c}_{\boldsymbol{i},\downarrow}|\mathrm{N}\rangle$$

$$(50)$$

which directly yields the result in Eq. (47).

### B. String-based pairing mechanism

Finally we discuss implications of our results for the binding energies $E_{\mathrm{bdg}}$ defined in Eq. (1). Within the effective string model introduced in this article, we calculate the one-hole spinon-chargon energy $E_1$ and compare it to the energy $E_2$ of two holes bound by a string. Throughout we assume that the underlying Hamiltonian is of $t - J_z$ type, with pure Ising interactions in the background. As in the two-hole case we ignore (Trugman-) loop effects [13] or self-interactions of the strings [52].

For more complicated microscopic models, such as the $SU(2)$-invariant $t - J$ model, corrections to the binding energy must be expected. While our quantitative predictions in this case are of limited use, they nevertheless allow us to identify relevant competing processes that tend to support or prevent pairs of holes from forming a tightly bound state.

### 1. Binding energy in $t - J_z$ model

In Fig. 11 we show the negative binding energy $-E_{\mathrm{bdg}}$ in units of $t$, plotted as a function of $(J_z/t)^{2/3}$. When $-E_{\mathrm{bdg}} < 0$, the bound state of two holes connected by a string is energetically favorable and we predict pairing. This is the case for bosonic, distinguishable and fermionic holes when $J_z$ is sufficiently large. Indeed, in the $t - J_z$ model, the asymptotic binding energy in the tight-binding limit $J_z \gg t$ is $E_{\mathrm{bdg}} = J_z/2$ (see discussion below). Within our approximations this value is closely approached for fermionic holes already at $J_z = t$, where bosonic and distinguishable holes are predicted to be significantly more strongly bound. As shown in Appendix B, see Fig. 19, all hole-types approach the asymptotic value $J_z/2$ when $J_z/t \gtrsim 10$.

Our effective theory clearly shows a significant increase of the binding energy for bosonic or distinguishable holes, as compared to the case with fermionic statistics. We attribute this to the frustrating effect of fermionic minus signs identified first by Trugman [13], which are picked up when two indistinguishable fermionic holes are exchanged and the string is completely reversed.

Bosonic and distinguishable holes have identical binding energies, which is correctly predicted by the asymmetric approximation. This is due to the bosonic nature of bound states within the asymmetric approximation at

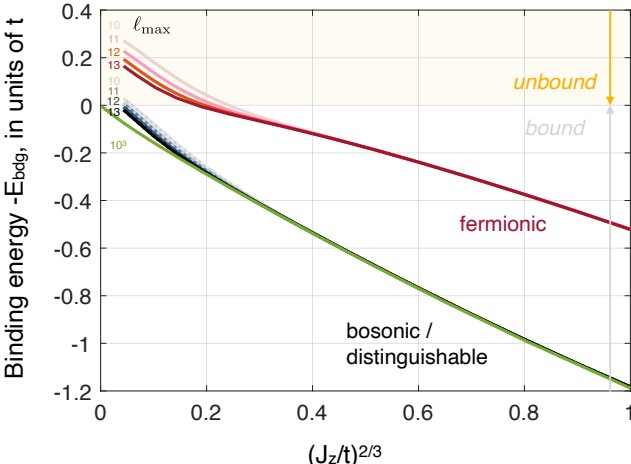

FIG. 11. Binding energy from the effective theory, predicted for the $t - J_z$ model. We used the truncated basis including all $m_4 = 0, .., 3$ and $m_3^{(1)} = 0, 1, 2$ sectors, and string lengths up to $\ell_{\max}$ (as indicated in the plot). The string potential was calculated for an Ising background. Around $\boldsymbol{k} = 0$ there is no coupling to rotational states for the indistinguishable holes and a linear string theory allows to work with much longer maximum string lenghts $\ell_{\max} = 10^3$.

the dispersion minimum around $\boldsymbol{k} = 0$ which we discussed above. Because all rotational quantum numbers $\boldsymbol{m}$ are conserved within the linear string approximation introduced in Sec. III A 3, we can solve the case with distinguishable holes at $\boldsymbol{k} = 0$ for much larger cut-offs $\ell_{\max}$ within the asymmetric approximation (which is exact in this case) and test how strongly $E_{\text{bdg}}$ depends on the maximum string length $\ell_{\max}$. To this end in Fig. 11 we compare results formally including all $m_4$ and $m_3^{(1)}$ sectors with variable $\ell_{\max}$ up to 13 to exact results at $\boldsymbol{m} = 0$ with $\ell_{\max} = 10^3$. We find that for $(J_z/t)^{2/3} > 0.2$, corresponding to $J_z/t > 0.09$ the finite-string length results with $\ell_{\max} = 13$ are reasonably well converged.

For fermionic holes in the strong coupling regime $J_z \ll t$, we find that the binding energy approaches zero within our effective theory. In this regime larger values of $\ell_{\max}$ would be required to reach convergence, and thus we cannot draw a final conclusion whether holes are bound or unbound for $(J_z/t)^{2/3} > 0.2$ (i.e. $J_z/t > 0.09$).

### 2. Pairing mechanisms for holes

For pairing to be energetically favorable one requires an intricate balance of different microscopic effects influencing the one- and two-hole ground state energies $E_{1,2}$ entering the expression for the binding energy (1). To shed more light on the underlying binding mechanism, we highlight various microscopic processes that either support or prevent pairing.

*Tight-binding limit: $J_{(z)} \gg t$.–* In this case, hole motion can be ignored and the energetically most fa-

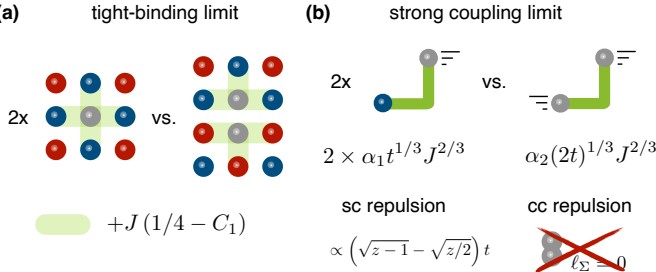

FIG. 12. Discussion of the underlying pairing mechanisms. (a) In the tight-binding limit $J_{(z)} \gg t$, kinetic contributions can be ignored. In this case the distortion of the Néel background can be minimized while also gaining maximal energy from the nearest-neighbor attraction term in the $t - J_{(z)}$ model. (b) In the strong coupling limit the hole's kinetic energy dominates, leading to a powerful pairing mechanism when one hole retraces the string of the other. This mechanism is supported by an effective spinon-charge (sc) repulsion with a geometric origin [28] and suppressed by the hard-core chargon-chargon (cc) repulsion.

vorable locations of the holes can be determined, see Fig. 12 (a). On one hand, the nearest-neighbor attraction $-J_{(z)}/4 \hat{n}_{\boldsymbol{i}} \hat{n}_{\boldsymbol{j}}$ in the microscopic models (2), (3) favors tightly bound hole pairs. In addition, each hole breaks up antiferromagnetic bonds which contributed an energy $J_{(z)} C_1$ in the undoped ground state. Ignoring any backaction of the localized holes on their spin environment, one finds that neighboring holes feel an attractive binding energy $J_{(z)}(1/4 - C_1)$. In the Ising case this result is exact, yielding $E_{\text{bdg}} = J_z/2$ as claimed above.

*Strong coupling limit: $J_z \ll t$.–* In this case, the kinetic energy of the holes plays an important role. As long as string formation is energetically favorable and the Nagaoka effect is suppressed [22], the leading order kinetic energy per hole is $E_{\text{kin}}^{(0)} = -2t_{\text{eff}}$. Here $t_{\text{eff}} = \sqrt{z - 1}t$ is the effective hopping constant between string states on the Bethe lattice, and $z$ is the coordination number of the underlying physical lattice [11, 27]. This asymptotic zero-point contribution to the energy per hole is identical for spinon-chargon and chargon-chargon pairs, hence canceling exactly in the binding energy $E_{\text{bdg}}$ [42].

Since the string potential is assumed to be linear in the string length within our theory, see Sec. III A 3, the next-to-leading order contribution to the energy $E_n$ takes the universal form [42]

$$E_n = -2t_{\text{eff}} + \alpha_n (nt)^{1/3} J_{(z)}^{2/3} + \mathcal{O}(J_{(z)}) \qquad (51)$$

for $n = 1, 2$-holes states. The factor of $n = 2$ renormalizing the tunneling in the two-hole state reflects the reduced mass $m_{\text{red}} = m/2 = 1/(2t)$ describing the relative motion of the hole pair attached to the string. This result can be directly obtained from Eq. (30); or see Ref. [42] for a simplified derivation. The pre-factors $\alpha_n > 0$ are non-universal constants, generally depending on $n$, which are determined by the details of the string potential.

From Eq. (51) we obtain a powerful binding mechanism if we assume $\alpha_1 \approx \alpha_2 = \alpha$. In this case,

$$E_{\rm bdg} = \left(2 - 2^{1/3}\right)\alpha\, t^{1/3} J_{(z)}^{2/3} > 0 \qquad (52)$$

asymptotically when $t \gg J_z$, see also Ref. [42]. This binding energy scales with a non-trivial power of $t^{1/3}$ and thus can easily exceed $J_{(z)}$ deep in the strong coupling limit. Indeed, for bosonic and distinguishable holes in Fig. 11 we confirm the scaling predicted by Eq. (52) for $J_z/t \to 0$, where $E_{\rm bdg}/t \propto (J_z/t)^{2/3}$ as expected. Plotted on a linear scale over $J_z/t$, see Fig. 19 in Appendix B, we find a clear curvature of $E_{\rm bdg}/t$ deep in the strong coupling regime.

In general, the coefficients $\alpha_1$ and $\alpha_2$ are not identical. For example, as discussed above the fermionic statistics lead to an effective repulsion between holes, which can cause an increase of $\alpha_2$. Indeed, for fermionic holes at strong coupling due to finite-size effects we cannot identify a clear asymptotic behavior of the binding energy in Fig. 11. Over a significant range of values $J_z/t$ our results are consistent with $E_{\rm bdg} \propto J_z$, see also Fig. 19 in Appendix B, which corresponds to $\alpha_1/\alpha_2 \approx 2^{-2/3}$. In the following we discuss two further effects which influence the coefficients $\alpha_n$, as summarized in Fig. 12 (b).

*Hard-core repulsion of holes.–* A single hole forming a spinon-chargon bound state can realize the zero-length $\ell_\Sigma = 0$ string state, corresponding to a spinon and a chargon on the same lattice site. In our effective description of two holes, we explicitly excluded such states to account for the hard-core nature of the holes. This effective chargon-chargon repulsion generally leads to a larger value of $\alpha_2 > \alpha_1$, suppressing the tendency to pairing.

In Ref. [42] we proposed another microscopic model with two layers, which allows to realize [43] distinguishable hole pairs with opposite layer indices. In this system, the hard-core constraint of the holes is effectively removed and one can realize $\alpha_1 = \alpha_2$. By comparison to numerical DMRG simulations we confirmed the strong pairing expected by Eq. (52) in that model due to the hole's gain in kinetic energy [42]. This demonstrates the importance of finding other mechanisms which allow to overcome the strong repulsion of hard-core holes, either by increasing $\alpha_1$ or decreasing $\alpha_2$.

*Spinon-chargon repulsion in dimensions $d > 1$.–* One such mechanism is the geometric spinon-chargon repulsion at strong coupling $J_{(z)} \ll t$ described in Ref. [28]. This effect is due to a decreased zero-point kinetic energy around the string-length $\ell_\Sigma = 0$ state, which leads to a localized repulsive interaction of strength $\propto t$. This in turn causes $\alpha_1$ to increase and approach $\alpha_2$, mimicking the hard-core repulsion of two holes and thus supporting pairing in the strong-coupling regime.

Quantitatively the spinon-chargon repulsion can be best understood by considering the ro-vibrational ground state with $\boldsymbol{m} = 0$. As shown in Sec. III (see also Appendix A), the latter can be described by a hopping problem on a semi-infinite lattice $\ell_\Sigma = 0, 1, 2, ...$ with an effective hopping strength $t_{\rm eff} = \sqrt{z-1}t$ in the bulk, and

$t_0 = \sqrt{z}t$ between $|\ell_\Sigma = 0\rangle$ and $|\ell_\Sigma = 1\rangle$. To capture edge effects around $\ell_\Sigma = 0$, this problem can be mapped to the even-parity sector of a hopping problem on an infinite one-dimensional lattice $\ell = ..., -2, -1, 0, 1, 2, ...$ [28], with tunneling strength $t_1 = \sqrt{z/2}t$ between $|\ell = \pm1\rangle$ and $|\ell = 0\rangle$ and $t_{\rm eff}$ otherwise. When $t \gg J_{(z)}$ this yields an effective repulsive interaction around $\ell = 0$ with strength

$$g_{\rm sc}(z) = \left(\sqrt{z-1} - \sqrt{z/2}\right)t, \qquad (53)$$

with $g_{\rm sc}(z) > 0$ for $z > 2$, i.e. in dimensions $d \geq 2$. In one dimension, the effect is absent, indicating a tendency to avoid pairing in $d = 1$.

*Spinon dynamics.–* Finally we note that spinon dynamics can also contribute to the energy of the one-hole spinon-chargon state. In the $t - J_z$ case the effective spinon hopping is due to Trugman loops [13] which lead to a negligibly small spinon kinetic energy a small fraction of $J_z$ [27, 50] for any ratio $t/J_z$. In the $t - J$ case, the spinon hopping can lead to a further reduction of the spinon-chargon energy on the order $J$, which provides another mechanism suppressing pairing for experimentally realistic ratios of $t/J$.

## V. SUMMARY AND OUTLOOK

In summary, we have studied an effective model of a pair of mobile dopants bound together by a strongly confining string. Our work is motivated by the physics of holes moving in an antiferromagnet, which is believed to be at the heart of many strongly-correlated electron systems. While the model applies most directly to mobile dopants in an Ising antiferromagnet, we believe it also has relevance upon including spin-flip terms or in entirely different settings such as in a strongly confined regime of lattice gauge theories with dynamical matter where the strings correspond to gauge fields. We also studied the effect of exchange statistics of the charge carriers, which allowed us to extend our model to Hubbard-Mott excitons with distinguishable dopants (a doublon bound to a hole) or a theoretical scenario with bosonic spins featuring antiferromagnetic interactions.

To study the nature of the bound states predicted by the effective string model, we analyzed their ro-vibrational excitation spectra with full momentum resolution. This allowed us not only to reveal their binding energies, but more importantly gives access to the pair dispersion relations. Of the latter we revealed two types of bands: One, strongly dispersive bands, where one dopant retraces the string of the other giving rise to high-mobility of strongly bound pairs, even in situations where an isolated single dopant would feature a strongly renormalized mass. Second, flat bands, where the dopants still form a bound state with a strongly fluctuating average distance, but where destructive quantum interference effects suppress any center-of-mass motion of

the pair. Such flat-band states of pairs have previously been mentioned [14] but, to our knowledge, never been analyzed in greater detail.

The main results of our work can be summarized as follows. For fermionic holes doped into a Néel state, most directly related to the problem of high-$T_c$ superconductivity in cuprates, we reveal a low-lying flat band corresponding to a pair with $d$-wave symmetry. Only in the direct vicinity of the $\Gamma$-point in the Brillouin zone we found a strongly dispersive paired state with $s$-wave symmetry. While the fate of these states upon including fully $SU(2)$-invariant spin-exchange terms remains to be clarified, the prospect of forming flat, or nearly-flat $d$-wave pairs at low energies suggests that they may be relevant for understanding competing phases featuring charge localization, such as stripe [56] and pair density wave [57] states. Thereby our analysis sheds new light on the question about the origin of superconductivity and possible connections to other systems, such as twisted bilayer graphene, believed to feature nearly flat bands.

For bosonic and distinguishable pairs of dopants, most relevant to Hubbard-Mott excitons, we found a stronger tendency towards pairing than in the fermionic case. We believe this is due to the frustrating effect of the fermionic exchange sign on the charge's kinetic energy [13]. In the spectra of bosonic and distinguishable dopants we revealed a strongly dispersive lowest-energy state with $s$-wave pairing symmetry, at energies significantly below the first flat-band bound state. We analyzed the underlying pairing mechanism in some detail, and predict a universal scaling of the binding energy $|E_{\rm bdg}| \propto t^{1/3} J^{2/3}$ for bosons and distinguishable charges in the strong-coupling regime. For fermions, in contrast, we revealed a reduced tendency towards pairing and due to finite-size effects the asymptotic behavior of $E_{\rm bdg}$ at strong couplings remains to be clarified.

Our work sets the stage for future extensions. In particular, it will be important to include the effects of quantum fluctuations, i.e. spin-flip terms, in the surrounding antiferromagnet. Starting from the effective string model developed here, the so-called generalized $1/S$-expansion technique [27] can be used to capture such effects as additional polaronic dressing of the bound states with magnons. For the strongly dispersive bands we found here, only relatively weak renormalization by magnon dressing can be expected to occur due to the large recoil energy associated with magnon emission from a pair. On the other hand, the flat band states with a localized center-of-mass are expected to be more strongly renormalized by magnons, and we anticipate that a weak dispersion can be induced. This picture is consistent with our recent DMRG results [39].

Another noteworthy assumption we have made is to focus on tightly bound pairs of charges connected by a string. On the other hand, and as described in the text, individual charges can also form spinon-chargon bound states. By including magnon dressing of the latter, long-range effective interactions between these one-hole states

can also be induced. These could lead to other types of bound states, with a character entirely different from the tightly-bound chargon-chargon pairs described in this article. Exploring such states, and their connection to the tightly-bound string states discussed here, will be a worthy future endeavor.

## ACKNOWLEDGEMENTS

We thank M. Hafezi, I. Bloch, M. Greiner, L. Vidmar, U. Schollwöck, T. Hilker, S. Hirthe and L. Homeier for fruitful discussions, and L. Homeier for critical reading of our manuscript. This research was funded by the Deutsche Forschungsgemeinschaft (DFG, German Research Foundation) under Germany's Excellence Strategy – EXC-2111 – 390814868, by the European Research Council (ERC) under the European Union's Horizon 2020 research and innovation programm (Grant Agreement no 948141 — ERC Starting Grant SimUc-Quam), by the NSF through a grant for the Institute for Theoretical Atomic, Molecular, and Optical Physics at Harvard University and the Smithsonian Astrophysical Observatory. ED acknowledges support from the ARO grant number W911NF-20-1-0163.

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

## Appendix A: Theory of spinon-chargon bound states

In this Appendix [58] we present a string-based model of spinon-chargon bound states in a 2D AFM described by the fermionic $t - J$ Hamiltonian. We extend the formalism introduced in the main text to include NNN spinon hopping terms. Although the following calculation largely follows the treatment of parton pairs presented in the main text, we will provide a self-contained discussion and derivation.

The formalism we develop includes the momentum dependence of the spinon-chargon bound states, thus improving previous theoretical models based on geometric strings [27, 41] by including spinon dynamics beyond the strong-coupling limit. Our predictions based on the theoretical model presented below have previously been shown to yield very good agreement with fully numerically obtained rotational spectra of individual holes, see Ref. [29].

### 1. Model

We include strong spin-charge correlations by working in the effective Hilbert space obtained by the geometric string construction [27]. The corresponding basis states are labeled by the position of the spinon $\boldsymbol{x}_{\mathrm{s}}$ in the 2D square lattice, and the string $\Sigma$ along which spins are displaced. The chargon (spinon) is located at the end (beginning) of the string $\Sigma$. Here $\Sigma = \{\boldsymbol{e}_1, \boldsymbol{e}_2, ..., \boldsymbol{e}_\ell\}$ denotes a sequence of steps $\boldsymbol{e}_n = \pm \boldsymbol{e}_{x,y}$ without direct re-tracing, i.e. $\boldsymbol{e}_{n+1} \neq -\boldsymbol{e}_n$; more conveniently, string states $\Sigma$ can be represented by the sites of a Bethe-lattice, or Cayley tree, with coordination number $z = 4$.

Every spinon-chargon basis state $|\boldsymbol{x}_{\mathrm{s}}, \Sigma\rangle$ has a microscopic representation by a quantum state $|\psi(\boldsymbol{x}_{\mathrm{s}}, \Sigma)\rangle$ in the $t - J$ model, defined by

$$|\psi(\boldsymbol{x}_{\mathrm{s}}, \Sigma)\rangle = \hat{G}_\Sigma \sum_\sigma \hat{c}_{\boldsymbol{x}_{\mathrm{s}}, \sigma} |\Psi_0\rangle, \qquad (A1)$$

where $\hat{c}_{\boldsymbol{j}, \sigma}$ is a microscopic fermion operator at site $\boldsymbol{j}$ with spin $\sigma$. Further, $|\Psi_0\rangle$ denotes the ground state of the undoped Heisenberg model and the operator $\hat{G}_\Sigma$ displaces all spins along the string $\Sigma$ while simultaneously moving the hole [41].

The geometric string states $|\psi(\boldsymbol{x}_{\mathrm{s}}, \Sigma)\rangle$ form an overcomplete and non-orthogonal basis of the one-hole $t - J$ Hilbert space. However, to a good approximation we may assume that most of the relevant string states are orthonormal [41]. This motivates our definition of the effective spinon-chargon Hilbertspace, which is spanned by the set of orthonormal basis states $|\boldsymbol{x}_{\mathrm{s}}, \Sigma\rangle$ with

$$\langle \boldsymbol{x}_{\mathrm{s}}, \Sigma | \boldsymbol{x}'_{\mathrm{s}}, \Sigma' \rangle = \delta_{\Sigma, \Sigma'} \delta_{\boldsymbol{x}_{\mathrm{s}}, \boldsymbol{x}'_{\mathrm{s}}}. \qquad (A2)$$

Note that our choice of the Hilbert space is similar to the non-retracing string approximation proposed by Brinkman and Rice [12], but in addition we include the spinon degrees of freedom $\boldsymbol{x}_{\mathrm{s}}$.

The effective Hamiltonian $\hat{\mathcal{H}}$ describing spinon-chargon bound states can be obtained by calculating matrix elements of the microscopic $t - J$ Hamiltonian, $\langle \psi(\boldsymbol{x}_{\mathrm{s}}, \Sigma) | \hat{\mathcal{H}}_{tJ} | \psi(\boldsymbol{x}'_{\mathrm{s}}, \Sigma') \rangle$. The hopping part $\propto t$ yields tunnelings between nearest-neighbor sites $\langle \Sigma, \Sigma' \rangle$ on the Bethe lattice:

$$\hat{\mathcal{H}}_t^{\mathrm{c}} = -t \sum_{\langle \Sigma, \Sigma' \rangle} |\Sigma\rangle\langle\Sigma'| + \mathrm{h.c.}, \qquad (A3)$$

independent of the spinon position.

The spin-exchange terms $\propto J_\perp$ in $\hat{\mathcal{H}}_{tJ}$ introduce spinon dynamics. Assuming for simplicity that $|\Psi_0\rangle$ is given by a classical Néel state along $z$, we obtain next-nearest neighbor tunneling of the spinon [41] which is correlated with a re-organization of the string:

$$\hat{\mathcal{H}}_J^{\mathrm{s}} = \frac{J_\perp}{2} \sum_{\boldsymbol{x}_{\mathrm{s}}, \Sigma} \sum_{(\boldsymbol{e}_2, \boldsymbol{e}_1)}{}' |\boldsymbol{x}_{\mathrm{s}} + \boldsymbol{e}_2 + \boldsymbol{e}_1\rangle\langle\boldsymbol{x}_{\mathrm{s}}| \otimes |\Sigma_{\boldsymbol{e}_2, \boldsymbol{e}_1}\rangle\langle\Sigma|. \quad (A4)$$

Here the sum $\Sigma'$ is over consecutive links $\boldsymbol{e}_2 \neq -\boldsymbol{e}_1$ for which the spins on sites $\boldsymbol{x}_s + \boldsymbol{e}_1$ and $\boldsymbol{x}_s + \boldsymbol{e}_1 + \boldsymbol{e}_2$ are anti-aligned for the given string configuration $\Sigma$; the string $\Sigma_{\boldsymbol{e}_2,\boldsymbol{e}_1}$ is obtained by adding or removing the first two steps $\boldsymbol{e}_1$ and $\boldsymbol{e}_2$ from the original string $\Sigma$ (see Ref. [41] for a discussion).

The remaining spin-exchange terms $\propto J_z, J_\perp$ in $\hat{\mathcal{H}}_{tJ}$ give rise to spinon-chargon interactions,

$$\hat{\mathcal{H}}_J^{sc} = \sum_\Sigma V_\Sigma |\Sigma\rangle\langle\Sigma| \approx \sum_\Sigma V(\ell_\Sigma)|\Sigma\rangle\langle\Sigma|. \tag{A5}$$

In the second step we assume that the string potential $V_\Sigma$ depends only on the length of the string $\ell_\Sigma$ (linear string approximation). We calculate $V(\ell_\Sigma)$ by considering straight strings, which yields

$$V(\ell_\Sigma) = \frac{dE}{d\ell}\ell_\Sigma + g_0\delta_{\ell_\Sigma,0} + \mu_h. \tag{A6}$$

The linear string tension $dE/d\ell$, $g_0$ and $\mu_h$ can be expressed in terms of the correlations in the undoped parent AFM, see [41].

## 2. Symmetries and quantum numbers

The effective Hamiltonian defined in the spinon-chargon Hilbert space,

$$\hat{\mathcal{H}} = \hat{\mathcal{H}}_t^c + \hat{\mathcal{H}}_J^s + \hat{\mathcal{H}}_J^{sc}, \tag{A7}$$

is manifestly translationally invariant, $[\hat{\mathcal{H}}, \hat{T}_\mu] = 0$. The translation operator leaves the string state unchanged and shifts the spinon position, $\hat{T}_\mu = e^{-i\hat{\boldsymbol{P}}_s \cdot \boldsymbol{e}_\mu}$ where $\mu = x, y$. Hence, we can label eigenstates by the spinon momentum $\boldsymbol{k}_s$ in the lattice.

Furthermore, the underlying $t - J$ model has an exact $\hat{C}_4$ discrete rotational symmetry, which carries over to the effective Hamiltonian: $[\hat{\mathcal{H}}, \hat{C}_4] = 0$. In the new Hilbert space, $\hat{C}_4$ rotates the chargon position $\boldsymbol{x}$ (the string configuration $\Sigma$) around the origin in the lattice (of the Bethe lattice). Hence, eigenstates can also be labeled by $m_4 = 0, 1, 2, 3$ corresponding to eigenvalues $\exp(im_4\pi/2)$ of $\hat{C}_4$.

In general, $\hat{C}_4$ and $\hat{T}_\mu$ do not commute and we cannot simultaneously assign linear and angular momentum quantum numbers. Exceptions require momenta $\boldsymbol{k}_s$ for which $\hat{\mathcal{H}}(\boldsymbol{C}_4\boldsymbol{k}_s) = \hat{\mathcal{H}}(\boldsymbol{k}_s)$. In particular, this is the case for $C_4$-invariant momenta (C4IM), i.e. when the rotated momentum $\boldsymbol{C}_4\boldsymbol{k}_s$ is equivalent to $\boldsymbol{k}_s$ modulo the reciprocal lattice vector,

$$\boldsymbol{C}_4\boldsymbol{k}_s^{C4IM} \equiv \boldsymbol{k}_s^{C4IM} \mathrm{mod}\boldsymbol{G}. \tag{A8}$$

The reciprocal lattice vectors depend on the unit cell. In the square lattice with a one-site unit cell, the resulting C4IM are: $\boldsymbol{k}_s^{C4IM} = (0,0)$ and $(\pi,\pi)$. In the AFM phase,

where the sub-lattice symmetry is spontaneously broken, the C4IM of the magnetic unit-cell are

$$\boldsymbol{k}_s^{C4IM} = (0,0), (\pi,0), (0,\pi), (\pi,\pi). \tag{A9}$$

While the anti-nodal points are $C_4$ invariant, we emphasize that the nodal points $(\pm\pi/2, \pm\pi/2)$, where the ground state of the magnetic polarons is located, are *not* $C_4$ invariant.

If we make the linear string approximation in Eq. (A5), the system at strong coupling $t \gg J_\perp$ has a series of additional discrete $\hat{C}_3$ rotational symmetries around the nodes of the Bethe lattice different from the origin; see main text or Ref. [27] for a detailed discussion of the resulting $m_3$ eigenvalues.

## 3. Solution within linear string approximation

To solve the effective Hamiltonian (A7) for the linear string potential (A6), we start from the following basis,

$$|\boldsymbol{k}_s, \ell_\Sigma, \boldsymbol{m}\rangle = \frac{1}{\sqrt{V}}\sum_{\boldsymbol{x}_s} e^{i\boldsymbol{k}_s \cdot \boldsymbol{x}_s}|\boldsymbol{x}_s, \ell_\Sigma, \boldsymbol{m}\rangle. \tag{A10}$$

Here $\ell_\Sigma$ and $\boldsymbol{m} = (m_4, m_3^{(1)}, m_3^{(2)}, ...)$ denote the string length and angular momenta on the Bethe lattice to label string configurations $\Sigma$ [27]; $V = L^2$ is the area of the physical lattice.

We truncate the basis by neglecting higher angular momenta beyond $m_3 \equiv m_3^{(1)}$; i.e. we consider only states with $m_3^{(n)} = 0$ for $n \geq 2$, see Fig. 13. This is motivated by the strong coupling result ($J_\perp = 0$) that non-zero rotational quantum numbers $m_3^{(n)} \neq 0$ lead to higher spinon-chargon interaction energies [27]: This is easily understood by noting that non-zero values $m_3^{(n)} \neq 0$ correspond to eigenvectors which are superpositions of strings with lengths $\geq n + 1$, see Ref. [27].

Since $\boldsymbol{k}_s$ is conserved, the effective Hamiltonian in the truncated basis is fully defined by its matrix elements

$$H_{\ell',m_4',m_3';\ell,m_4,m_3}(\boldsymbol{k}_s) = \langle \boldsymbol{k}_s, \ell', \underbrace{m_4', m_3'}_{\boldsymbol{m}'}|\hat{\mathcal{H}}|\boldsymbol{k}_s, \ell, \underbrace{m_4, m_3}_{\boldsymbol{m}}\rangle. \tag{A11}$$

The latter are relatively easy to calculate if we make use of symmetries: The chargon hopping $\hat{\mathcal{H}}_t^c$ conserves all angular momenta $\boldsymbol{m}$ on the Bethe lattice [27] and couples only $\ell$ and $\ell \pm 1$. It is sufficient to consider one direction – we choose $\ell \to \ell' = \ell - 1$ – since the other follows from the condition that $\hat{\mathcal{H}}$ is hermitian. From $\hat{\mathcal{H}}_t^c$ we obtain:

$$H_{\ell-1,\boldsymbol{m}';\ell,\boldsymbol{m}}^c = -t\delta_{\boldsymbol{m}',\boldsymbol{m}} \times \begin{cases} \sqrt{z-1}, & \ell \geq 2 \\ \sqrt{z}, & \ell = 1 \end{cases} \tag{A12}$$

independent of $\boldsymbol{k}_s$, where $z = 4$ is the coordination number of the lattice.

For the spinon hopping Eq. (A4) only transitions between $\ell \to \ell \pm 2$ are allowed and the angular momenta

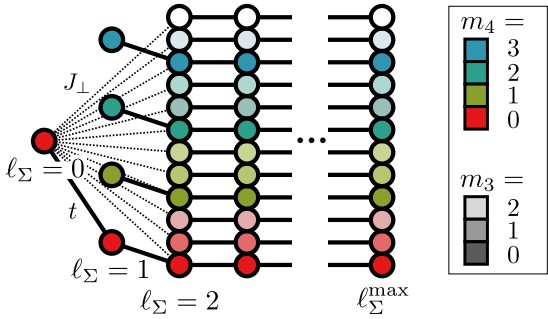

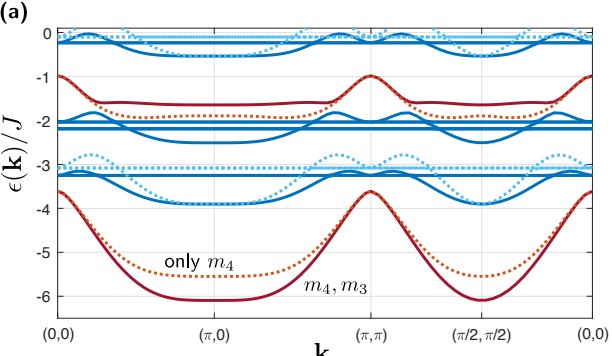

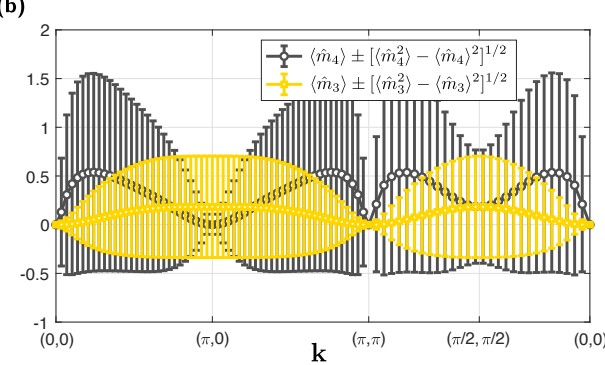

FIG. 13. Truncated rotational string basis. For fixed center-of-mass momentum $\boldsymbol{k}_\mathrm{s}$ of the spinon-chargon pair, we work with a truncated string basis. At string lengths $\ell_\Sigma = 0, 1$ all states are included, and labeled by their discrete $C_4$ and $C_3$ angular momentum quantum numbers $m_4$ and $m_3 = m_3^{(1)}$ in the Bethe lattice. For longer strings with length $\geq 3$ we only include $m_4$ and $m_3$ excitations and set higher $m_3^{(>1)} = 0$. The generally allowed matrix elements for hole (solid lines, $t$) and spinon (dotted lines, $J_\perp$) hopping are indicated (in the case of the spinon, only those involving $\ell_\Sigma = 0$, for clarity).

$\boldsymbol{m}$ can change in this process. A full calculation for our lattice with $z = 4$ yields

$$H_{\ell-2,\boldsymbol{m}';\ell,\boldsymbol{m}}^\mathrm{s} = J_\perp \times \begin{cases} \frac{1}{\sqrt{3}}[\Lambda_{\boldsymbol{m}}^\mathrm{s}\delta_{\boldsymbol{m}',\boldsymbol{0}} - \Phi_{\boldsymbol{m}',\boldsymbol{m}}^\mathrm{s}], & \ell > 2 \\ \frac{1}{2}\delta_{\boldsymbol{m}',\boldsymbol{0}}\Lambda_{\boldsymbol{m}}^\mathrm{s}, & \ell = 2 \end{cases}.$$
(A13)

Here we first defined

$$\Lambda_{\boldsymbol{m}}^\mathrm{s}(\boldsymbol{k}_\mathrm{s}) = \frac{1}{2\sqrt{3}}\sum_{\nu=0}^{3}\sum_{\nu'=1}^{3} e^{i\nu m_4 \frac{\pi}{2}} e^{i\nu' m_3 \frac{2\pi}{3}} e^{-i\boldsymbol{k}_\mathrm{s}\cdot\boldsymbol{e}_{\nu',\nu}},$$
(A14)

with $\boldsymbol{e}_{\nu',\nu}$ re-tracing the first two string segments, starting to count at the spinon position; $\nu\pi/2$ denotes the angle of the first string segment relative to the $x$-axis (i.e. $\nu = 0, 1, 2, 3$) and $(\nu'-2)\pi/2$ denotes the angle of the second string segment relative to the first (i.e. $\nu' = 1, 2, 3$). In complex notation (i.e. real and imaginary parts of $\epsilon_{\nu',\nu} \in \mathbb{C}$ represent the $x$ and $y$ components of $\boldsymbol{e}_{\nu',\nu} \in \mathbb{R}$) it holds:

$$\epsilon_{\nu',\nu} = e^{i\nu\frac{\pi}{2}} + e^{i\nu\frac{\pi}{2}} e^{i(\nu'-2)\frac{\pi}{2}}.$$
(A15)

We further defined for longer strings:

$$\Phi_{\boldsymbol{m}',\boldsymbol{m}}^\mathrm{s}(\boldsymbol{k}_\mathrm{s}) = \frac{1}{4}\sum_{\nu=0}^{3} e^{i(m_4-m_4')\nu\frac{\pi}{2}}\chi_{\boldsymbol{m}}^\mathrm{s}(\boldsymbol{k}_\mathrm{s},\nu),$$
(A16)

with:

$$\chi_{\boldsymbol{m}}^\mathrm{s}(\boldsymbol{k}_\mathrm{s},\nu) = \frac{1}{2\sqrt{3}}\sum_{\nu'=1}^{3} e^{i\nu' m_3 \frac{2\pi}{3}} e^{-im_4\nu'\frac{\pi}{2}} e^{-i\boldsymbol{k}_\mathrm{s}\cdot\boldsymbol{e}_{\nu',\nu-\nu'}}.$$
(A17)

By diagonalizing the effective Hamiltonian $H^\mathrm{s}(\boldsymbol{k}_\mathrm{s})$ in the truncated basis (Fig. 13), we obtain all low-energy spinon-chargon bound states and their dispersion relations. The ground state is adiabatically connected to

FIG. 14. String model results. (a) We show the lowest energy bands predicted by the spinon-chargon model for $t/J = 3$ and $J_\perp = J$. A cut along high-symmetry directions in the Brillouin zone is shown. The full solid lines correspond to the full truncated basis; the light dotted lines correspond to a further truncated basis with only $m_4 \neq 0$ included. Colors indicate how states connect to rotational (blue) and vibrational (red) bands at $\boldsymbol{k} = \boldsymbol{0}$. (b) We show the expectation value and variance (error bars) of the rotational eigenvalues $\hat{m}_4$ (black) and $\hat{m}_3$ (yellow) for the ground state of the spinon-chargon model; parameters as in (a).

$\boldsymbol{m} = \boldsymbol{0}$ without rotational excitations at $\boldsymbol{k}_\mathrm{s}^\mathrm{C4IM}$; within our simplified spinon model Eq. (A4), it has a degenerate energy minimum at the edge of the magnetic Brillouin zone including nodal and anti-nodal points. This dispersion closely resembles the ground state magnetic polaron dispersion, although it misses the small energy splitting between nodal and anti-nodal points [41]. The low-energy excited states have non-trivial rotational quantum numbers, and their dispersion relations feature a richer structure. The spinon hopping causes quantum interference effects between rotationally excited states which are degenerate in the absence of spinon hopping.

At the $\boldsymbol{k}_\mathrm{s}^\mathrm{C4IM}$ of the magnetic Brillouin zone, see Eq. (A9), the Hamiltonian $H_{\ell',\boldsymbol{m}';\ell,\boldsymbol{m}}(\boldsymbol{k}_\mathrm{s}^\mathrm{C4IM})$ is block-diagonal, with blocks labeled by $m_4 = 0, 1, 2, 3$. At $\boldsymbol{k}_\mathrm{s} = 0$ the block with $m_4 = 0$ also conserves the $m_3$ quantum numbers, since $\chi_{m_4=0,m_3}^\mathrm{s}(0,\nu) \propto \delta_{m_3,0}$. One further finds that $\chi_{m_4\neq0,m_3=0}^\mathrm{s}(0,\nu)$ is equal for all $m_4 = 1, 2, 3$, which means that the three lowest-order rotational states with $m_4 \neq 0$, $m_3 = 0$ are degenerate at $\boldsymbol{k}_\mathrm{s} = 0$.

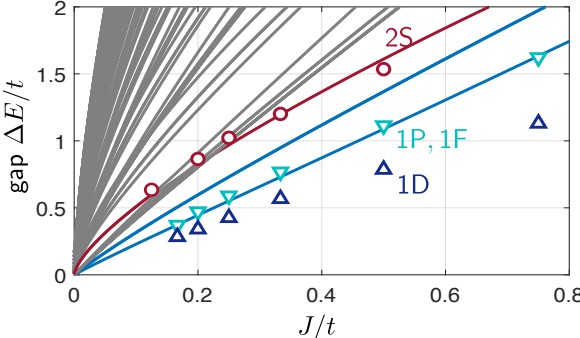

FIG. 15. Meson Regge trajectories from the spinon-chargon model introduced here. We calculate the excitation energy gaps $\Delta E$ from the ground state from the spinon-chargon model at the nodal point, $\boldsymbol{k} = (\pi/2, \pi/2)$, solid lines. These predictions are compared to our numerical DMRG results (data points), see Ref. [29]. The lowest excitations can be identified as rotational (blue) and vibrational (red) by the dependence of their energy gap on $J/t$. Higher excited states (gray) show similar, though less pronounced behavior.

### 4. Results

Now we apply the spinon-chargon model introduced above to calculate ro-vibrational eigenstates. In Fig. 14 (a) we show all low-energy spinon-chargon eigenstates along high-symmetry cuts through the Brillouin zone. Although well-defined $C_4$ rotational quantum numbers $m_4$ can only be assigned at C4IM, we can still clearly identify sets of states which are adiabatically connected to the rotational or vibrational states at the C4IM. The ro-vibrational ground state is non-degenerate, lowest red band in Fig. 14 (a). Then we find a band consisting of three rotational states, which correspond to the non-trivial rotational states $m_4 \neq 0$ at the C4IM, lowest blue band in Fig. 14 (a). At $\boldsymbol{k} = \boldsymbol{0} \equiv \boldsymbol{\pi} \bmod \boldsymbol{G}$, the latter are exactly degenerate.

To estimate the effect of higher rotational excitations with $m_3^{(n)} \neq 0$, in Fig. 14 (a) we also compare our results to model calculations where we truncate the basis further and include only $m_4$ states while setting all $m_3^{(n)} = 0$. For the lowest-lying vibrational and rotational states, we observe a modest shift to lower energies when $m_3^{(1)} \neq 0$ states are included. The ground state at $\boldsymbol{k} = \boldsymbol{0} \equiv \boldsymbol{\pi} \bmod \boldsymbol{G}$ is an exception: As described in the previous section, all $m_3^{(n)} = 0$ quantum numbers are explicitly conserved at this point in the model, and the ground state energy is exactly obtained.

We find that the third band of states (solid blue) we identify in Fig. 14 (a) consists of eight states, some of which are degenerate. This band is only obtained if $m_3^{(1)}$ excitations are included. Indeed, this number of states was predicted at strong coupling for higher-order rotational excitations with $m_3^{(1)} = 1, 2$ (each of those has four distinct $m_4$ states) [27]. Away from the C4IM, the

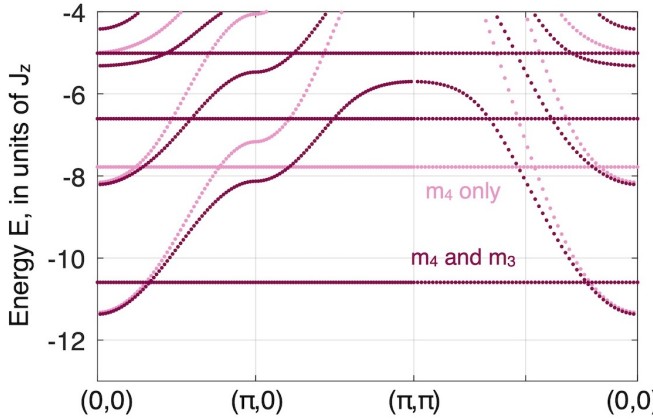

FIG. 16. Two-hole spectrum as in Fig. 6, showing only data for fermionic holes. We compare results from the truncated basis with all $m_4$ and $m_3^{(1)}$ sectors (dark dots) to results from a further reduced basis including only $m_4$ sectors but $m_3^{(n)} = 0$ for all $n$ (light dots). In both cases we used a maximum string length $\ell_{\max} = 11$, and we considered $t/J_z = 3$ with a string potential for an Ising background.

non-trivial $m_3^{(1)} \neq 0$ excitations weakly hybridize with the purely vibrationally excited state (2S) and we observe small avoided crossings. The counting suggests that the energetically highest shown three states correspond to the rotationally excited, $m_4 \neq 0$, versions of the 2S state, with a vibrational quantum number $n = 2$.

In Fig. 14 (b) we calculate the expectation values $\langle \hat{m}_4 \rangle$ and $\langle \hat{m}_3 \rangle$ for the ground state. The error bars denote the variance. As expected, we find that $m_4 = 0$ is a good quantum number (zero variance) at C4IM of the magnetic Brillouin zone. At $\boldsymbol{k} = \boldsymbol{0} \equiv \boldsymbol{\pi} \bmod \boldsymbol{G}$, even $m_3 = 0$ is a good quantum number with zero fluctuations. All other momenta show some hybridization of $m_4$ and $m_3$ quantum numbers.

In Fig. 15 we apply the string model to calculate Regge trajectories [29] at the nodal point. We find that the energy gap $\Delta E$ to the three lowest-lying excitations scales linearly with $J$, the hallmark signature expected of rotational states. We also compare our results to numerical DMRG calculations from Ref. [29]. Without any free fit parameters, we find that the energy gap to the first vibrational excitation (2S) is accurately predicted by the spinon-chargon model.

Because of the hybridization of different $m_3$ and $m_4$ states with each other, the model predicts a splitting between different states from the lowest rotational excitation. While the overall scale of this splitting is correctly predicted, we find numerically from DMRG a smaller than expected energy gap to the rotational states. As the DMRG data, the model predicts a non-degenerate lower rotational line and a two-fold degenerate higher rotational line. However, we found that the distribution of spectral weight in the model differs from the DMRG results [29].

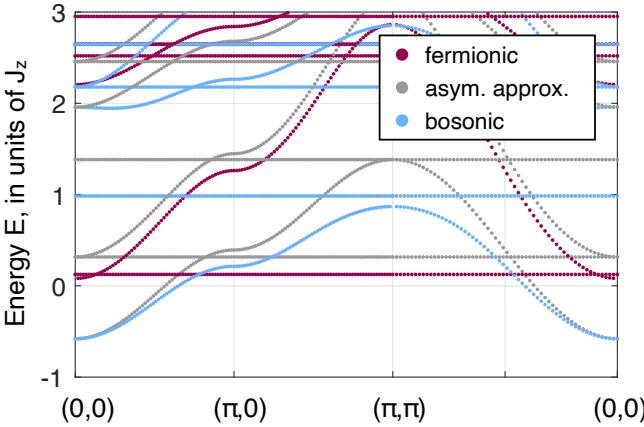

FIG. 17. Two-hole spectrum as in Fig. 6, but for $t/J_z = 1$. We used a maximum string length $\ell_{\max} = 11$, and assumed a string potential for an Ising background.

## Appendix B: Additional results for two holes

In this Appendix we present additional numerical checks and results for two holes that broaden our understanding of the truncated basis method developed in the main text.

In Fig. 16 we demonstrate that the inclusion of $m_3^{(1)}$ states in the truncated basis leads to a significant shift of some eigenenergies, in particular of the flat bands and for large momenta around $\boldsymbol{k} = (\pi, \pi)$. In contrast, around the ground state $\boldsymbol{k} = \boldsymbol{0}$, the inclusion of $m_3^{(1)}$ has no or only little effect. We checked and obtained similar behavior for bosonic and distinguishable holes.

In Fig. 17 we show eigenenergies of the two-hole Hamiltonian for different statistics as in Fig. 6 of the main text, but for smaller $t/J_z = 1$. Our results are qualitatively unchanged, but in the fermionic case we observe a closer

energetic competition of the lowest-energy $\boldsymbol{k} = 0$ state with the lowest fermionic flat-band state.

In Fig. 19 we show the binding energy, calculated from the effective string theory in a $t - J_z$ model. We used the same data as in Fig. 11 but included larger values of $J_z/t$ on the linear $x$-axis.

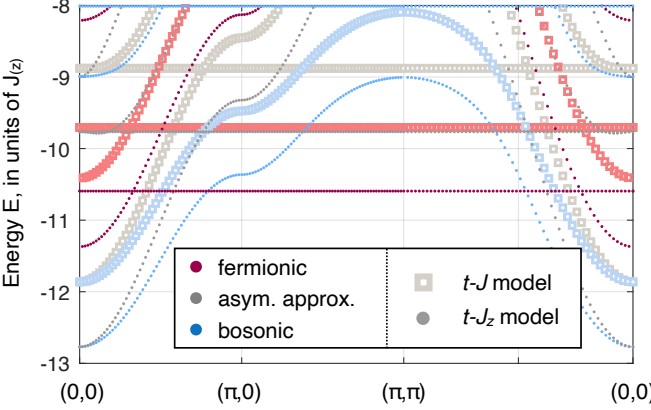

FIG. 18. Two-hole spectrum as in Fig. 6, comparing the previous $t - J_z$ results to spectral lines calculated for a string-tension Eq. (16) calculated for a weakly doped $t - J$ model (with $J_2 = 0$). We assumed $t/J_z = 3$ ($t/J = 3$) and used a maximum string length $\ell_{\max} = 11$.

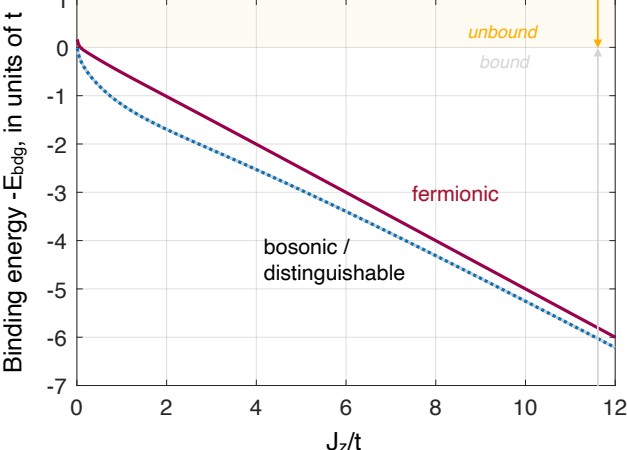

FIG. 19. Negative binding energy $-E_{\mathrm{bdg}}$ as shown in Fig. 11, but plotted linearly over $J_z/t$ and including larger values of $J_z > t$. We used a maximum string length of $\ell_{\max} = 13$.