# Peer review of "Pairing of holes by confining strings in antiferromagnets"

_SciPost Physics_

## Round 1 · Referee Report · Anonymous (Referee 1) · 2022-10-13

Report

The authors reconsider a long-standing problem,
the properties of two propagating impurities
(holes, doubly occupied sites) in a Neél background.
The authors propose an approximate theory based on a
simplified string potential. The following remarks are
not necessarily in order of relevance.

(a)
Strings of flipped spins break into two parts when
self-intersecting. Self-intersection is neglected
by the authors on the basis that the relative number
of self-intersecting strings is small. This
is true, the argument is nevertheless a fallacy,
at least on a formal level.

Relative to its last direction of movement, a hole can
move right/straight/left (R/S/L). Strings are hence
sequences of R/S/L tokens, such as ...RSSLLS...

A string will self-intersect if the patterns
..RRR.. or ..LLL..
occur. The probability for this to happen is
2/3**3 = 2/27 = 7.4%, as listed in Table 1.

The probability that strings do not self-interact
decays exponentially with the length of the string,
roughly as (1-0.074)**(length)

This is relevant because R/S/L hoppings are associated
with distinct spin configurations, which makes the
propagation of holes equivalent to that of a classical
particle, which is also stated by the authors. Strings
ceases to exist on a formal basis once they self-intersects
for the first time.

Neglecting self-intersection is hence OK for short, but
not for long strings. The authors work in the limit t >> J,
which implies a weak confining potential and hence long
strings. This consideration suggests therefore that the
approach presented is void, on a formal basis, being
based on long, but not self-intersecting strings. The
effect of string intersections on an effective confining
potential needs to be worked out (see next point). It is
a bit disturbing that the authors did not discuss this issue.

(b)
An equivalent problem concerns the confining potential.
The patterns ..LL.. or ..RR..
lead to adjacent string elements and hence to a correction
to the confining potential, here with respect to the linear
approximation used by the authors. The probability for this
to occur for every two steps is substantial, 2/9 = 22.2%

Given that string self-interactions are attractive, LL and RR
token patterns will occur dynamically even with a larger
probability. The same holds for the arguments in (a). The
linear simplification used by the authors is hence
questionable. It is a bit strange that this issue is
not mentioned in the manuscript.

(c)
The authors claim that their approach works both for
the t-J_z and the t-J model, as long as t>>J_z,
respectively t>>J. This claim seems blatantly wrong.
The probability that J_xy fluctuations disrupt the
string increases exponentially with string length,
the presumed regime of validity of the proposed
approach. Strings cease to exist altogether, at least
in their naive form. It is worrisome that the authors
did not mention this well-known problem.

(d)
The transformation from the Hubbard to the t-J model
leads to an intra-sublattice correlated hopping term,
let's call it here T_2. The prefactor is J. It allows
for coherent intra-sublattice propagation, which
contrasts qualitatively with the bare NN hopping. No strings
are generated by T_2. Without a word, the authors disregard
T_2, keeping only the bare J term. This can be valid
for suitable ion-trap experiments, but most probably not
for solid-state applications. In view of the amusing
phrasing:

"The system most closely related to our effective
theory is constituted by the 2D t-J_z model on a
square lattice."

the authors may argue that their theory is any case
only somewhat remotely related to real-world physics.
But leaving out a term that would add qualitative
new features, and possibly invalidate the entire
formalism, needs supporting arguments.

(e)
The two end-impurities of a string are governed by
identical dynamics. It would have hence been intuitive
to use double-sided stacks for the encoding of strings.
The authors opted instead for an asymmetric formulation
that needs to be symmetrized in a second step. Is there
a rational for the choice of an asymmetric encoding?
  • validity: -
  • significance: -
  • originality: -
  • clarity: -
  • formatting: -
  • grammar: -

Author:  Fabian Grusdt  on 2023-01-06  [id 3213]

(in reply to Report 1 on 2022-10-13)

Please find our complete reply in the attached pdf.

Attachment:

RefereeReply-v2_O5eqKsD.pdf

---

## Round 1 · Referee Report · Anonymous (Referee 2) · 2022-11-27

Report

See attached pdf file.

Attachment

  • validity: -
  • significance: -
  • originality: -
  • clarity: -
  • formatting: -
  • grammar: -

Author:  Fabian Grusdt  on 2023-01-06  [id 3214]

(in reply to Report 2 on 2022-11-27)

Please find a completely reply to all referee reports in the attached pdf.

Attachment:

RefereeReply-v2_0jH1kyD.pdf

---

## Editorial Decision

resubmitted